# Sensory and Physicochemical Characterization of Sourdough Bread Prepared with a Coconut Water Kefir Starter

**DOI:** 10.3390/foods9091165

**Published:** 2020-08-24

**Authors:** Mansi Limbad, Noemi Gutierrez Maddox, Nazimah Hamid, Kevin Kantono

**Affiliations:** Department of Food Science and Microbiology, Auckland University of Technology, 34, Saint Paul Street, Auckland 1010, New Zealand; noemi.gutierrezmaddox@aut.ac.nz (N.G.M.); nazimah.hamid@aut.ac.nz (N.H.); kevin.kantono@aut.ac.nz (K.K.)

**Keywords:** sourdough, sensory, *L. fermentum*, *L. plantarum*, dietary fibre, coconut water kefir, amino acids, organic acids

## Abstract

There is a recognized need for formulating functional food products using selected lactic acid bacteria (LAB) starter cultures from various sources such as kefir, yoghurt or kombucha that have health benefits. The principle objective of this study was to investigate the use of a coconut water kefir-based fermentation starter culture using *Lactobacillus fermentum* and *Lactobacillus plantarum* to develop a sourdough bread. Check-all-that-apply (CATA) sensory profiling was used in this study to evaluate the sensory profile of sourdough breads that varied with culture type, culture concentrations, with and without added yeast, and with fermentation for 18 and 24 h. Based on correspondence analysis (CA) of the CATA results, bread samples with positive sensory attributes were chosen for further physicochemical analysis. Physicochemical analyses (texture, proximate composition, shelf life, carboxylic acid analysis and amino acid analysis) were carried out on breads formulated with starter culture concentrations of 8.30 log CFU/mL of *L. fermentum*, 4.90 log CFU/mL of *L. fermentum* and 9.60 log CFU/mL of *L. plantarum*, each fermented for 24 h without baker’s yeast. The bread sample that was formulated with a coconut water kefir (CWK) starter culture containing 9.60 log CFU/mL of *L. plantarum*, without dry yeast and fermented for 24 h, had significantly higher values for almost all amino acids and a lower protein content compared to samples formulated using CWK cultures containing 8.30 log CFU/mL of *L. fermentum* and 4.90 log CFU/mL of *L. fermentum*, both without dry yeast and fermented for 24 h. The bread sample formulated with CWK starter culture containing 9.60 log CFU/mL of *L. plantarum*, without dry yeast and fermented for 24 h, also produced significant quantities of organic acids (pyruvic acid, acetic acid, lactic acid and succinic acid). These changes in the physicochemical properties can improve overall bread quality in terms of flavor, shelf life, texture and nutritional value.

## 1. Introduction

Sourdough has been used in bread production for dough leavening [1], but also helps extend shelf life, improve nutritional properties, increase bioactive compound contents and improve bread flavor [2,3,4]. Sourdough is a complex ecosystem in which lactic acid bacteria (LAB) and yeast interact together and with bread ingredients, depending on the process parameters [5]. The endogenous factors in flours and technological parameters employed for sourdough processing play a key role in influencing the microbial communities [6].

Traditional sourdough breads have health-promoting attributes that include the reduction of risks associated with colorectal cancer, cardiovascular disorders, diabetes and obesity. The sourdoughs used in bread production usually contain a wide range of probiotics which bestow positive effects on health [7]. Various functional and nutritional features of sourdough fermentation, such as its application for salt reduction, reducing irritable bowel syndrome (IBS), synthesis or release of bioactive compounds and the metabolism of phenolic compounds, as well as its ability to lower glycemic index, increase mineral bioavailability and decrease the gluten content, have been proven [8,9,10,11,12,13].

Sensory analysis is an important tool which helps in developing new food products to determine the level of consumer satisfaction. The Check-All-That-Apply (CATA) method can be used to understand consumer perception of a product effectively compared to other descriptive analysis methods, which require highly trained panelists. In this method, a list of sensory attributes are presented to the panelists, asking them to select all the attributes which relate to the product appearance, flavor, taste, smell and/or texture [14,15].

Evidence suggests that it is important to use LAB strains with high phytase and glutamic acid content because they can improve the nutritional and functional properties of sourdough bread. The phytase enzyme sequentially dephosphorylates phytic acid to products of lower chelating capacity and higher solubility, and in turn, increases mineral absorption [16]. Evidence suggests that both *L. plantarum* and *L. fermentum* are capable of producing high extracellular phytase activity [17,18]. The phytase enzyme has been the subject of much systematic investigation, since it plays an important role in increasing the nutritional and functional properties of the bread, and provides sensory advantages to the product by increasing the total free amino acid content [19]. Existing research also recognizes the critical role played by glutamic acid in enhancing the flavor and functional properties of the sourdough. Glutamic acid is a precursor to important amino acids such as proline and arginine, as well as some bioactive precursors like glutathione and γ-aminobutyric acid (GABA). GABA has been classified as a bioactive component in foods and pharmaceuticals [20,21,22].

Fermentation is the major route for volatile formation in sourdough and bread crumb. It produces mainly acids, alcohols, aldehydes, esters and ketones [23]. Acidification (especially the formation of acetic acid) is the main factor that enhances pungent flavor [24] and diminishes fresh flavor [25]. However, acidification has also been shown to be a key factor in the induction of proteolysis, the main factor enhancing, e.g., roasted flavor, during sourdough fermentation [26]. LAB species contribute to flavor due to their ability to bring about acidification by the production of lactic and acetic acid. Additional flavor compounds can be generated due to conversion of amino acids like phenylalanine (sweet), isoleucine (acidic), asparagine, glycine, serine and alanine (vinegar/sour) to aldehydes and ketones [27,28].

The liquid endosperm of coconut water is of cytoplasmic origin, which fills the cavity within the coconut [29]. Coconut water contains almost all components of the vitamin B group (except for B6 and B12), minerals, proteins, sugars, amino acids, magnesium, vitamin C, potassium, and growth factors such as auxins, cytokinins and gibberellins, which makes it biologically favorable for human nutrition as well as for the growth of microorganisms, and an ideal medium for kefir fermentation [30,31,32,33,34]. This isotonic drink is very low in fat and also contains optimal amounts of RNA phosphorous, which play an active role in the transport of amino acids and respiratory metabolism in living cells [35]. This study was aimed at developing a sourdough bread produced from a novel sourdough fermented with CWK cultures containing two selected LAB strains (*Lactobacillus fermentum* and *Lactobacillus plantarum*) that had high production of phytase and glutamic acid. Sensory analysis using the CATA sensory profiling method was carried out on the baked sourdough breads. Bread samples with desirable sensory properties were then subjected to further physicochemical testing.

## 2. Materials and Methods

### 2.1. Microorganisms Used to Formulate the Sourdough

Pure cultures of two LAB isolates with the ability to produce high concentrations of glutamic acid and phytase enzyme were supplemented at known cell concentrations into the sourdough. The isolates, *L. fermentum* and *L. plantarum*, were isolated, identified and purified from CWK. Identification was carried out from the fermented CWK via DNA extraction using a PowerFood DNA Isolation kit (Mo Bio Laboratories, Carlsbad, CA, USA) and the Sanger’s method for DNA sequencing. The raw data (DNA sequences) were received and analyzed using Geneious prime Bioinformatics Software Pro 5.6 (Geneious, Auckland, New Zealand), and identified using the National Center for Biotechnology Information’s Basic local alignment search tool (NCBI) BLAST. Each isolate, stored with glycerol (20%) at −80 °C, was revived in De Man, Rogosa and Sharpe agar (MRS) broth (100 mL) and incubated for 48 h at 30 °C in a 5% CO_2_ incubator. The broth cultures were spun down in a GYROZEN centrifuge, model 1580R (Bio-strategy, Auckland, New Zealand) at 4000× *g* for 15 min to obtain a cell pellet. The cell pellet for each isolate was washed with sterile, deionized water and resuspended in fresh, sterile, deionized water. The cells were washed three times to remove all traces of the MRS broth. The cell suspensions were prepared fresh on the same day the dough was produced.

The concentration of each isolate was measured using a spectrophotometer (Ultrospec 2100 pro UV/visible spectrophotometer). The absorbance reading was converted to a unit of log CFU/mL by reference to a standard curve drawn between the absorbance values (600 nm) and known colony units (log CFU/mL). Baker’s yeast (Edmond’s instant dry yeast, New Zealand) used in the dough was purchased from Countdown, Auckland. A CWK sourdough bread was formulated using *Lactobacillus fermentum* and *Lactobacillus plantarum*. The isolates from CWK with high glutamic acid and phytase producing ability were used. As *L. fermentum* and *L. plantarum* produced significant concentrations of glutamic acid and phytase enzyme, both a high and a low cell concentration of these isolates were used to develop a CWK fermented sourdough bread (data not shown). Other factors included the addition of dry yeast (at 1.16 g) and fermentation time (18 h and 24 h).

### 2.2. Production and Formulation of Sourdough Breads Using Factorial Analysis

A D-Optimal Design (Table 1) was used to generate the formulations of sourdough breads using high and low concentrations of *L. plantarum* and *L. fermentum* as variables, along with the addition of baker’s yeast as an independent variable [36]. The D-Optimal design was generated using the Unscrambler 10.1 (Camo Analytics AS, Oslo, Norway) software. The experimental concentrations for Isolate 1 (*L. fermentum*) were 4.90 Log CFU/mL (low concentration) and 8.30 Log CFU/mL (high concentration; and for Isolate 2 were 5.0 Log CFU/mL (low concentration) and 9.60 Log CFU/mL (high concentration). The product was formulated with or without addition of 1.16 g of baker’s yeast and fermented for either 18 or 24 h. Different fermentation times, i.e., 18 h and 24 h, were used for fermenting the sourdough bread samples to determine the best condition to achieve high acidity and high dough volume. Baker’s yeast was only added to samples fermented for 18 h for the purpose of speeding up fermentation.

CWK was prepared using 300 mL of UFC coconut water (manufactured by Universal Food Public Company Limited, Thailand, purchased at Countdown, Auckland City), 1.5 g/L of kefir starter (Body Ecology™, Auckland, New Zealand) and 12.00 g/L of sucrose (Food grade, Chelsea sugar limited, Auckland, New Zealand). Coconut water kefir was allowed to ferment for up to 48 h at 30 °C in a LabServ incubator (Thermo Fisher, Auckland, New Zealand) before being used in bread preparation. The fermented mixture of coconut water kefir used to prepare the sourdough contained 12 g/L of sucrose as it produced the highest cell counts for LAB and yeast (data not shown), and faster sugar utilization that resulted in low concentrations of residual sugar at the end of 96 h of fermentation.

High-grade flour (Home brand, Countdown, Auckland, New Zealand) and Canola oil (Gilmours, Auckland, New Zealand) were used to make the sourdough bread. No water was used in preparation of the sourdoughs; only CWK was used. All the sourdough breads were made by mixing salt, dry yeast, 48 h fermented CWK and inoculum as specified in Table 1 to form a dough. The dough was prepared at 27 ± 2 °C, using a Breville BBM100WHT Baker’s Oven Breadmaker (Auckland, New Zealand). This dough was then kept in the proofer (Unox Linemiss XL193-B, Hospitality Supply, Auckland, New Zealand) to ferment at 30 °C and 70% humidity. The oven (Piron, Auckland, New Zealand) was preheated at 175 °C for 30 min. The doughs were then baked at 175 °C for 25 min, with 10% relative humidity, in an oven that. Three experimental replicates for each formulation were tested.

### 2.3. Sensory Analysis of Breads

#### 2.3.1. Ethics Statement and Location

The Auckland University of Technology Ethics Committee approved the sensory study (AUTEC ethic application 16/340) carried out in this research. The panelists gave written and informed consent prior to the commencement of the study. Sensory testing took place at the Auckland University of Technology Sensory Suite, Auckland, New Zealand. Sensory sessions were approximately 60 min in duration.

#### 2.3.2. Sensory Attributes

The sensory attributes evaluated in this study are listed in Table 2. Each food reference consisted of either 20 mL of solution or 25.00 g of solid reference food, which was served at room temperature (22 °C) in a 30 mL black plastic cup or a paper plate. The samples were coded with three-digit random numbers.

#### 2.3.3. Panelists

Forty-nine semi-trained panelists (a mix of males and females) between 21 and 39 years of age participated in this study (Mean age = 30, standard deviation = 9.81). They were recruited online through an advertisement posted on social networking services (i.e., Facebook and Instagram), or on university bulletin boards, and were compensated for their participation. None of the panelists was a smoker or suffered from eating disorders or other health problems associated with food that may interfere with hearing or understanding the instructions, or during gustation. Once all the bread samples were prepared, they were subjected to CATA sensory testing within 12 h of baking. All data were collected between the hours of 10:00 am and 2:00 pm, and participants were directed not to eat anything 2 h prior to the commencement of sensory testing.

#### 2.3.4. Panelist Training

Panelists were invited to attend a sensory training session for Check-All-That-Apply (CATA) for product sensory assessment. They were trained over three 5-h sessions, with a total training time of 15 h. Sensory attributes associated with CWK sourdough bread were generated, as shown in Table 2. Participants were asked to familiarize themselves with the definitions of the attributes. The panelists were then instructed to select multiple attributes that they perceived when consuming the bread samples.

#### 2.3.5. Check-All-That-Apply (CATA)

The procedure used for the CATA method was carried out as described by Castura, Antúnez et al. and Baker, Castura and Ross [37,38]. Assessors were instructed to review the attributes prior to sample evaluation, and to familiarize themselves with the sensory attributes and their definitions (Table 2). Assessors were asked to check the attributes that described the sensory characteristics of samples during evaluation.

#### 2.3.6. Tasting Conditions

The William Latin square design [45] was used to present the bread samples and ensure that the order effects were balanced out. Three-digit random numbers were used to code each product, which were cut in the dimensions of 1cm × 1cm × 1cm (length × breadth × height) and weight of 5.0 ± 0.5 g. The panelists were requested to rinse their palate with water and consume unsalted crackers during a 60-s break. The testing was carried out in a temperature-controlled room (18 °C), under white light in individual booths.

### 2.4. Nutritional Composition and Physicochemical Analysis of the Sourdough Bread

The sourdough bread samples B, D and F with desirable sensory characteristics based on CATA sensory testing were subjected to further nutritional and chemical analyses. Sample F was made from sourdough containing 9.60 CFU/mL of *L. plantarum*, without dry yeast, and was fermented for 24 h. Samples B and D were made from sourdough containing 8.30 CFU/mL and 4.90 CFU/mL of *L. fermentum*, respectively, without dry yeast and fermented for 24 h. Three experimental replicates for each formulation were tested.

#### 2.4.1. Ash

The ash content was determined by igniting the sample at 550 °C in an electric furnace according to the AOAC 923.03 (Official Method 923.03, 2005) method [46].

#### 2.4.2. Carbohydrate

The carbohydrate content was determined according to the 1.2.8 FSANZ Food Standards Code, by subtracting the values for moisture content, total fat, ash, dietary fiber and protein content from 100%.
Carbohydrate content = 100 − (moisture content + total fat + ash + dietary fiber + protein content)

#### 2.4.3. Dietary Fiber

Determination of dietary fiber was carried out according to the AOAC method 985.29 [46].

#### 2.4.4. Fat

Determination of fat content in breads was carried out using the AOAC acid hydrolysis method 922.06 [47].

#### 2.4.5. Moisture

Determination of moisture content was carried out according to the AOAC method 925.09 [47].

#### 2.4.6. Protein

The Kjeldahl method was used as outlined by AOAC 2001.14 [48].

#### 2.4.7. Total Titratable Acidity

The total titratable acidity was determined by the AOAC method 942.15 [47]. Briefly, 5 g of bread sample was homogenized, and then 50 mL of neutralized water was added. One mL of phenolphthalein was then added to this mixture and titrated against 0.1 M NaOH solution until the end point was reached (solution became pale). Titratable acidity was calculated as g/100g of the acid in the sample.

#### 2.4.8. Enumeration of Yeasts and Molds

The method used to enumerate the yeasts and molds for determining the shelf life of bread products was obtained from the Compendium of Methods for the Microbial Examinations of Foods, Chapter 21.51, American Public Health Association, 5th Edition (2015) [49]. Briefly, Dichloran Rose Bengal Chloramphenicol (DRBC) agar, which contained 100 µg of chloramphenicol (added to the agar before autoclaving), Acetic Dichloran Yeast extract Sucrose agar (ADYS), Malt Extract agar (ME), Dichloran glycerol (18% added glycerol) agar (D18) and Tryptone Glucose Yeast Extract agar (TGY), was used for this assay. All the agar media were autoclaved for 121 °C for 15 min and were poured in petri plates that were allowed to cool overnight at a temperature of 25 °C under a dark environment.

One g of bread sample was completely homogenized and 9 mL peptone water was added. This was further diluted 9-fold, and 0.1 mL of these dilutions were spread plated on DRBC, D18, ADYS, ME and TGY agar media. The plates were incubated for 5 days at 25 °C, and the colonies were counted and reported as CFU/g for each bread sample.

### 2.5. Texture Analysis

Texture Profile Analysis (TPA) was performed to determine sourdough bread texture in terms of hardness, springiness, cohesiveness, chewiness, gumminess and resilience. All samples were prepared and baked on the day of the test. Each bread sample was cut using a knife to obtain a cube (2 cm × 2 cm × 2 cm). The cube was taken from the middle of each loaf to evaluate the crumb texture. Analyses were performed using a Stable Micro Systems TA.XT plus Texture Analyzer equipped with a Film Support Rig (HDP/FSR) on a Heavy Duty Platform (HDP/90) with a 5 mm stainless steel probe (P/55) and a 49.03 N load cell. The texture analyzer was set to measure force in the compression mode with a pretest speed of 2.0 mm/s, a test speed of 1.0 mm/s and a posttest speed of 10.0 mm/s time 30 s; load cell: 50 kg; trigger force: auto-10 g. Target mode was set to a distance of 5 mm; the data acquisition rate was set at 500 pps.

### 2.6. Determination of Carboxylic Acids Using LC-MS

The bread samples in triplicate were first ground with liquid nitrogen. Then, a 50 ± 0.1 mg sample was weighed and added to ice-cold 50% MeOH containing 10 mg/L 4-chlorobenzoic acid (CBA). CBA was used as an internal standard. The mixture was centrifuged at 10,000× *g* for 5 min at 4 °C (Z216MK, HERMLE Labortechnik GmbH, Germany). The 2-hydrazinoquinoline (HQ) derivatization reaction was conducted by mixing 5 μL of the bread sample with 100 μL of acetonitrile solution containing 1 mM 2,2′-dipyridyl dipyridyl disulfide (DPDS), 1 mM triphenylphosphine (TPP) and 1 mM HQ. The reaction mixture was incubated at 60 °C for 60 min. The HQ derivatives in the reaction mixture were analyzed using Liquid Chromatography–Mass Spectrometry (LC-MS).

The carboxylic acids used as standards for this assay were lactic acid, succinic acid, acetic acid and pyruvic acid, prepared with 50% methanol and 20 mM sodium hydroxide. The LC-MS was an Agilent 1260 Series liquid chromatograph comprising a G1311B quaternary pump, G1329B thermostatted autosampler and a G1330B thermostatted column compartment (Agilent Technologies, Santa Clara, CA, USA). Mobile phase A was 0.1% formic acid in ultrapure water, and mobile phase B was 0.1% acetic acid in acetonitrile. A Phenomenex Kinetex EVO C18 column was used, measuring 2.1 × 150 mm with 1.7 µm diameter packing material, maintained at 25 °C. The chromatographic gradient was held for 0.1 min at 3% B and then ramped to 15% B at 6 min and 90% B at 9 min, before returning to 3% B at 11 min. The flow rate was 200 µL/min, the total run time was 16 min and the injection volume was 5 µL.

For detection, an Agilent 6420 triple quadrupole mass spectrometer fitted with an Agilent Multimode Ionization source was operated in positive electrospray mode. Derivatized carboxylic acid peak areas were normalized and quantified by reference to a dilution series of external standards prepared. Data was collected and processed using the Agilent Mass Hunter software.

### 2.7. Determination of Amino Acids Using LC-MS

One gram of bread sample, in triplicates, with a mass of 45 ± 0.1 mg, was added to 1.40 mL volume of 50% methanol containing 10 mg L^−1^ of 2,3,3,3-d4-alanine (d4A) in a microcentrifuge tube. The tubes were vortexed to obtain a homogenous mixture. This was followed by centrifugation at 10.000 rpm for 5 min at 4 °C (Z216MK, HERMLE Labortechnik GmbH, Wehingen, Germany). A 10 µL volume of extract was used to perform precolumn derivatization with 6-aminoquinolyl-N-hydroxysuccinimidyl carbamate, following a method adapted from Salazar et al. [50].

The LC-MS was an Agilent 1260 Series liquid chromatograph comprising a G1311B quaternary pump, G1329B thermostatted autosampler and a G1330B thermostatted column compartment (Agilent Technologies, Santa Clara, CA, USA). The mobile phase A was 0.6% formic acid in ultrapure water and mobile phase B was 0.1% formic acid in acetonitrile. A Phenomenex Kinetex EVO C18 column was used, measuring 2.1 × 150 mm with 1.7 µm diameter packing material, maintained at 25 °C. The chromatographic gradient was held for 1 min at 1.5% B and then ramped to 13% B at eight min, 17% B at 15 min and 80% B at 16 min, before returning to 1.5% B at 17.5 min. The flow rate was 300 µL/min, total run time was 28 min, and the injection volume was 5 µL. For detection, an Agilent 6420 triple quadrupole mass spectrometer fitted with an Agilent Multimode Ionization source was operated in positive electrospray mode. Optimum Multiple Reaction Monitoring (MRM) transitions were established using Agilent Mass Hunter Optimizer B06.00 software (Agilent Technologies, Santa Clara, CA, USA). Derivatized amino acid peak areas were normalized to the recovery of d4A and quantified by reference to a dilution series of external standards prepared from a commercial amino acid mix (Sigma product A9906, Sigma-Aldrich Pty. Ltd., Sydney, Australia). Data were collected and processed using the Agilent Mass Hunter software.

### 2.8. Statistical Analysis

The results from analyses of amino acid, carboxylic acid, texture, total titratable acidity (TTA), shelf life testing and proximate composition for the different bread samples were subjected to one-way ANOVA using XLSTAT version 2019.1.2 (AddinSoft Inc., New York, NY, USA). Tukey’s post hoc comparison test was further carried out when significant differences were found for ANOVA to indicate significant differences between means.

Principle coordinate analysis was carried out using XLSTAT (version 2019.1.2, AddinSoft Inc., New York, NY, USA) according to Anderson and Braak [51]. The influence of variables (types of isolates: 1 and 2, and the concentration of isolates 1 and 2) on coconut water kefir fermentation was studied by employing a full factorial design [52]. Runs were performed at random. The results were analyzed using the XLSTAT (version 2019.1.2, AddinSoft Inc., New York, NY, USA) software.

Canonical Variate Analysis (CVA) was carried out to evaluate the differences between sourdough samples based on the sensory attributes, as described by Jager et al. [53]. In addition, Hotelling-Lawley trace Multivariate Analysis of Variance (MANOVA) tested for significant differences between each product loading at the 5% level. CVA was used in this study, as it can maximize the distances between products while minimizing residual variability [53].

For CATA analysis, the Cochran’s Q test [54] was carried out to identify significant differences among samples for each of the sensory terms. In this study, Cochran’s Q test was used to evaluate the differences between treatments (samples) with binary responses [55]. A correspondence analysis (CA) was performed on the frequency table from each experimental treatment considering chi-square distances. CA was used to visualize the frequency table as a generalization of Principal Component Analysis (PCA) for the attributes. The method projects the data into orthogonal components to maximize the sequential representation of the variation in the data [56,57]. Multidimensional Alignment (MDA) was applied to determine the cosine values between the bread products and sensory attributes originating in the CA of the CATA questions.

## 3. Results and Discussion

### 3.1. Sensory Analysis of Sourdough Breads Using Check-All-That-Apply (CATA)

Flavor perception for CWK- fermented sourdough bread samples produced using different concentrations of *Lactobacillus fermentum*, *Lactobacillus plantarum* with or without baker’s yeast was determined using CATA. Cochran’s Q test was carried out to identify any significant differences among samples for each of the descriptors in the CATA questionnaire [54]. The frequency of attributes across each sourdough sample was calculated. The results from the Cochran’s Q test showed significant differences for 36 out of 50 attributes among the eight samples (A–H) (*p* < 0.05) (Table 3).

The citation frequency of each attribute is summarized in Table 3. B, C, D, G and H breads were significantly more developed than breads A, E and F. ‘Developed’ is positively correlated with the breads, as reported by Plessas et al. and Katina et al. [58,59]. All breads were highly porous except for sample E. In fact, breads A, F and H were significantly more porous compared to all other bread samples. Porosity is related to the amount of carbon dioxide produced during the dough fermentation, which, in turn, is related to the presence of many nonvolatile and volatile compounds which are produced during dough fermentation. Porosity has been directly linked to improving elasticity, softness and springiness, which are the main parameters to determine the overall quality of the bread [60]. Porous (big holes) and soft bread could be attributed to the gas trapped in the gluten during fermentation that influences the viscoelastic properties of bread [61,62] by increasing gas retention during proofing and baking. Pores are formed due to the microbial activity of the LAB and yeast microorganisms during the fermentation of the dough, i.e., the production of carbon dioxide as the byproduct of the homo- or hetero- fermentation pathways [63,64,65,66].

The frequency of the ‘light’ attribute, regarding the appearance of the bread, was significantly high for all breads except C, E and F. Light brown color is a positive attribute of sourdough breads [66]. Bread crust undergoes Maillard reactions and caramelization, which are influenced by the quantity and quality of precursors in the dough, pH changes and the quantitative ratio of amino nitrogen to reducing sugar during baking [67,68]. Maillard reactions determine the browning intensities of the bread surface. It was observed that bread sample F had significantly high concentrations of important amino acids, as shown in Table 4, that participate in the Maillard reactions, such as lysine, glycine, tryptophan and tyrosine [69]. All the breads were significantly smoother in terms of appearance except bread B and E. Bread F had significantly higher shiny attribute compared to the rest of the samples. Amina, Ismail and Abdelkader [70] reported that the overall appearance of the bread is more likeable when the crust appears smooth, shiny, regularly colored (brown) and free of blisters.

Bread samples A, B, C, D, F, G and H were sour compared to bread E. The sour flavor in the breads could be attributed to the addition of LAB, which is brought about by acidification and proteolysis in the sourdough as a result of fermentation, as supported in results shown in Table 5 (Titratable Acidity) and Table 4 (Amino acids). Proteolysis during sour-dough fermentation produces amino acids, which are well-known precursors in bread flavor formation [71], while the acidification is the result of the formation of acids such as lactic acid and acetic acid, depending on the strain of LAB and yeast used for the fermentation of the dough [71]. It is well known that during dough fermentation, many acids like lactic acid and acetic acid are produced, which contribute to the perceived sourness of bread. Samples G, D and E had significantly higher frequency for freshness compared to the rest of the breads. Generally, the status of bread freshness as perceived by consumers depends on flavor (taste and aroma), appearance and crispness of crust, firmness of crumb, and bread volume.

Breads A, D and E had significantly higher frequency of spongy texture compared to bread sample C. The differences obtained between breads A, D, E and the rest of the bread samples could be related to the change in dough properties during and at the end of fermentation. This could be influenced by different structural components, such as gluten, and by changes in the pH value of the dough system [72]. Bread samples B and F had significantly firmer texture compared to sample D. This can be explained by the process of retrogradation, in which high concentrations of amino acids such as aspartic acid and glutamic acid (Table 4) may play an important role in increasing the retrogradation process. Aspartic acid and glutamic acid are both acidic amino acids, which can decrease the swelling power of the flour particles and cause increased gelatinization, syneresis and amylose leaching [73,74]. Samples F and B had significantly high concentrations of glutamic and aspartic acid, which could explain the firmer texture of these breads compared to sample D. Additionally, breads B and F that were firm did not have any additional baker’s yeast added to them during dough fermentation. A study carried out by Plessas, Pherson, Bekatorou, Nigam and Koutinas [59] reported that breads made with added baker’s yeast were softer. 

Breads C, G and H had significantly higher frequency of roasted flavor, compared to sample A. According to Katina et al. [75], the roasted flavor in sourdough correlates with the type and quantity of amino acids produced during fermentation. Samples A, F, G and H had a significantly higher frequency of yeasty flavor compared to breads B, C, D and E. Thiele, Gänzle and Vogel, [26] explained that sourdough breads which contain *S. cerevisiae* (commonly known as baker’s yeast) can be perceived as ‘yeasty’. Although yeasty odor is generally not associated with sourdough breads, the presence of *S. cerevisiae* in the fermented coconut water kefir or the added baker’s yeast to the sourdough could have contributed to breads A, F, G and H being perceived as more yeasty than the other breads.

Breads E, F, G and H were perceived as being significantly overcooked compared to breads A, B, C and D. ‘Overcooked’ is influenced by cooking temperature [76]. Interestingly, all the breads fermented with *L. plantarum* (breads E–H) were perceived as overcooked compared to all breads fermented with *L. fermentum* (breads A–D). Samples B, C, G and H had significantly hard texture compared to samples A, D, E and F. Generally, the breads are hard if there is less leavening activity and less acidity in the dough. The impact of sourdough on bread volume has been suggested to be principally due to enzymatic reactions taking place throughout the fermentation. These reactions are aided by the cereal enzymes which influence the metabolism of the LAB and yeast present in the dough during homo- or hetero- lactic fermentation, and produce byproducts such as carbon dioxide [75,77,78]. In addition, bread samples A, D and E had highly significant frequency for doughy, coarse, chewy and crumbly texture attributes compared to samples B, C, F, G and H.

#### Correspondence Analysis

A correspondence analysis (CA) was carried out to evaluate the CATA results and produce a bidimensional map (Figure 1) that explained 67.89% variance in the first two dimensions, as seen in Figure 1, with 43.98% and 23.91% variance explaining the first and second dimensions, respectively.

In addition, Multidimensional Alignment (MDA) was applied to determine the cosine values between the bread products and sensory attributes originating in the CA of the CATA questions. Table 6 highlights the attributes which are positively and negatively correlated with the bread samples using the first two dimensions of the CA bidimensional map.

As shown in Figure 1, the first dimension was positively correlated to samples C, H and G, and negatively correlated to samples E, D and A. On the other hand, the second dimension was positively correlated to samples F and B. The sensory map generated by CA further shows the relationship between the sensory attributes and the bread products. Bread samples C, G and H had high positive scores along dimension 1 (Figure 1). These samples were positively correlated with developed, shiny, greasy, dense crumb, beige, sweet, roast, supple, and hard attributes (Table 6). Attributes such as crumbly, doughy, blister and overcooked have been reported to be negatively correlated with sourdough bread [75].

Bread samples A and D had high negative scores along dimension 1 (Figure 1). These samples were positively correlated with the attributes of heterogenous, dark, airy crumb, buttery, spongy, dry, doughy, coarse, chewy and crumbly (Table 6). Flavor attributes such as nutty and buttery are positively related to the sourdough bread [75]. The heterogenous and light appearance of the bread is dependent on the baking temperature, and could be due to the flavor compounds formed during the Maillard reactions such as furans, pyrroles (thiophenes, thiophenones, thiapyrans and thiazolines), pyrazines, sulfur containing compounds and lipid degradation products such as alkanals, 2-alkenals and 2,4-alkadienals [69]. Bread sample E also had high negative loadings along dimension 1 (Figure 1). This sample was positively correlated with the attributes of airy crumb, spongy, fresh, doughy, coarse, chewy and crumbly (Table 6). Attributes such as doughy, crumbly and coarse have been negatively correlated with sourdough bread samples [75].

Bread samples B and F, that were not well explained along dimension 1, had high negative scores along dimension 2 (Figure 1). These samples were highly positively correlated with the attributes of light, blister, porous, small holes, big hole, nutty and moist (Table 6). The starter culture, ash content of the flour, fermentation temperature and dough yield have been reported to influence bread flavor. Furthermore, bread samples B and F had significantly high concentrations of amino acids such as lysine, glycine, tryptophan and arginine, which are involved in Maillard reactions that may produce important flavor compounds which positively contribute to the fresh and nutty attributes [69]. These flavor compounds are either volatile or nonvolatile, e.g., (1) 2,3-butanedione, 1-octen-3-one, phenylacetaldehyde, (Z)- 2-nonenal and 2-ethyl-3-methyl-pyrazine, that contribute to a sweet aroma; (2) 2-acetyl-1-pyrroline, that contributes to a roasted flavor; (3) 2-octenal, that contributes to nutty flavor, or (4) ethyl 2-methylpropanoate, that contributes to a fruity flavor [69]. Known positive attributes for sourdough breads include springiness, porous, sour, fresh and roasted [42,75,79,80,81].

From the present study, bread samples A, C, E, G, and H were eliminated from further studies due to their correlation with negative sensory attributes. Bread samples, B, D and F were chosen for further physicochemical characterization and GI determination, as they possessed positive attributes.

Starter culture used in bread samples A–H are shown below: A: 8.30 log CFU/mL of *L. fermentum*, 1.16 g of dry yeast and fermented for 18 h. B: 8.30 log CFU/mL of *L. fermentum*, without dry yeast and fermented for 24 h. C: 4.90 log CFU/mL of *L. fermentum*, 1.16 g of dry yeast and fermented for 18 h. D: 4.90 log CFU/mL of *L. fermentum*, without dry yeast and fermented for 24 h. E: 9.60 log CFU/mL of *L. plantarum*, 1.16 g of dry yeast and fermented for 18 h. F: 9.60 log CFU/mL of *L. plantarum*, without dry yeast and fermented for 24 h. G: 5.00 log CFU/mL of *L. plantarum*, 1.16 g of dry yeast and fermented for 18 h. H: 5.00 log CFU/mL of *L. plantarum*, without dry yeast and fermented for 24 h.

### 3.2. Proximate Composition and Texture Analysis

Proximate and texture analyses were carried out on sourdough bread samples B, D and F. The results are summarized in Table 5. In terms of proximate composition, there were no significant differences between the breads in terms of moisture, ash, carbohydrate, dietary fiber and fat content. Only results that showed significant differences in proximate composition and texture will be discussed.

#### 3.2.1. Protein Content

Only the protein content of bread sample D (8.50 ± 0.50/100 g) was significantly higher than that of sample F (8.2 ± 0.4/100 g) (*p* < 0.05). This could be explained by the fact that the proteolytic activity of sample F (composed of 9.60 log CFU/mL of *L. plantarum*) might be higher than sample D (composed of 4.90 log CFU/mL of *L. fermentum*). The decrease in the concentration of total protein may be due to the degradation of wheat flour proteins by the action of proteolytic enzymes (proteases) and LAB. This result is supported by the study carried out by Yin et al. [19] that showed that sourdough fermented with *L. plantarum* M616 resulted in an overall 8 g/100g decrease in the total protein content of bread after 24 h, which was attributed to the action of proteolytic enzymes (proteases) and LAB, depending on the fermentation time and pH. Rollán, De Angelis, Gobbetti and De Valdez, [82] further reported that *Lactobacillus plantarum* CRL 759 and 778, used to prepare sourdough, had a wide spectrum of peptidase activity during fermentation. They also reported that the degree of proteolysis was higher for dough starter with *L. plantarum* CRL 778, producing significant levels of basic, aliphatic and aromatic amino acids.

#### 3.2.2. Total Titratable Acidity (TTA)

The TTA of bread sample F (0.60 ± 0.10/100 g, *p* < 0.05) was significantly higher than samples B and D (0.4 100 g and 0.40 ± 0.10/100 g respectively, *p* < 0.05). Bello et al. [83] reported that sourdough bread fermented with *L. plantarum* had a significantly higher acidification rate, i.e., 0.23 ± 0.04/100 g (*p*-value < 0.05), compared to chemically acidified sourdoughs using acids such as citric, lactic and acetic acid. In addition, sample F had significantly higher concentrations of lactic acid, acetic acid, pyruvic acid and succinic acids (Table 5) compared to breads B and D.

#### 3.2.3. Shelf Life

A shelf life study of the three breads was carried out over two weeks by determining the CFU/g of yeast and molds. The yeast and mold counts were <10 CFU/g for samples F, B and D. The delay in yeast mold growth is likely associated with the presence of organic acids, as reported by Lynch et al. [84]. Sourdough-associated LAB produce many antimicrobial substances, such as organic acids, CO_2_, ethanol, hydrogen peroxide, diacetyl, fatty acids, phenyllactic acid, reuterin and fungicins [14,85].

#### 3.2.4. Texture Analysis

Texture results are shown in Table 5. Bread sample F had the highest hardness value (1137.10 ± 18.30, *p* < 0.05) compared to samples D and B (966.30 ± 20.20 and 937.30 ± 26.40 respectively, *p* < 0.05). Bread sample B had a significantly low value (76.70 ± 6.30, *p* < 0.05) for springiness compared to samples D and F (87.20 ± 2.40 and 87.90 ± 1.90 respectively, *p* < 0.05). The high values for hardness and springiness in sample F could be attributed to the high acidity in the sample. This is supported by the high values for total titratable acids shown in Table 5. The higher concentration of *L. plantarum* F could be responsible for the higher acidification, and thus, the higher hardness and springiness values. Bello et al. [83] reported a higher acidification rate for sourdough fermented with *L. plantarum* when compared to the chemically acidified sourdoughs. Hardness can also be attributed to the process of retrogradation, which is caused by the interaction of starch with amino acids such as aspartic acid and glutamic acid [86]. Sample F had high concentrations of both these amino acids, which could explain the high values for hardness in the sample (Table 4). Sample F had a significantly high resilience value (44.8 ± 3.5, *p* < 0.05) compared to samples D and B (34.90 ± 1.4 and 37.8 ± 0.6 respectively, *p* < 0.05).

### 3.3. Carboxylic Acids

Carboxylic acids, including lactic acid, acetic acid, pyruvic acid and succinic acid, were analyzed for the B, D and F CWK fermented bread samples using LCMS (Table 5). A study carried out by Corsetti [87] reported that the fermentation quotient is associated with the production of acids such as lactic and acetic acid and other volatile compounds during the sourdough fermentation. The production of acids and other compounds depends on microbial composition and activities. According to Table 5, sample F had the highest concentrations of lactic acid, acetic acid, succinic acid and pyruvic acid content, followed by samples B and D. Sample F had a high concentration (9.60 log CFU/mL of *L. plantarum* cells, and sample B had a high concentration (8.30 log CFU/mL) of *L. fermentum*. The high carboxylic acid contents in samples F and B were attributed to the high concentrations of LAB isolates. Hadaegh et al. [88] reported increased production of lactic and acetic acids in sourdoughs inoculated with high concentrations (30%) of starter cultures like *L. plantarum* jQ 301799, *L. casei* jQ412732 and/or *L. brevis* IBRC-M10790 compared to sourdoughs inoculated with 20% inoculum.

### 3.4. Amino Acids

Figure 2 shows the amino acid composition of B, F and D sourdough bread samples. A principal component analysis with bootstrapped confidence ellipses was carried out to better visualize the results. All the amino acid data points used for this graph clearly distinguished the samples along Factor 1 (F1). F1 and F2 explained a total of 92.3% variance, with F1 explaining 82.92% variance and F2 explaining 9.41% variance.

It can be observed that sample F had high positive scores along F1 and was associated with the highest loadings of almost all amino acids, except for arginine (Figure 2). Sample F had significantly high quantities of almost all amino acids (Table 4). Sample B had lower positive scores along F1 and was associated less amino acids than sample F. Samples F and B contained high concentrations of *L. plantarum* (9.60 log CFU/mL) and *L. fermentum* (8.30 log CFU/mL), respectively. Sample D, that had the highest negative score, was associated with the least amino acids and, in fact, was inoculated with the lowest concentration of *L. fermentum* (4.90 log CFU/mL).

The extent of the protein degradation, together with the proportion of LAB in the bread, are important parameters to consider, since they will have an impact on the final bread properties. A study carried out by Gobbetti et al. [89] reported that LAB starter culture (containing *Lactobacillus plantarum* DC400) added to sourdough at a high concentration, i.e., 10^7^ CFU/mL, had higher proteolytic activity and production of amino acids such as glutamic acids and alanine compared to other starters used for sourdough fermentation. This supports the results obtained in the present study, where sample F, containing *L. plantarum* at a concentration of 9.60 log CFU/mL, had the highest quantities of glutamic acid and alanine, histidine, hydroxy proline, arginine, serine, glycine, aspartic acid, threonine, proline, lysine, methionine, valine, tyrosine, isoleucine, leucine, phenylalanine and tryptophan. The increases in amino acid concentration may have been due to the greater inoculum concentration, which enhanced amino acid production [90,91], or amino acid excretion by cell autolysis during the 24 h period [92]. Increased proteolytic activities of *Lactobacillus plantarum* CRL 759 and 778 were observed in the sourdough, producing significant levels of basic, aliphatic and aromatic amino acids [82].

Starter culture in sourdough bread samples: F: 9.60 log CFU/mL of *L. plantarum*, without dry yeast and fermented for 24 h. B: 8.30 log CFU/mL of *L. fermentum*, without dry yeast and fermented for 24 h. D: 4.90 log CFU/mL of *L. fermentum*, without dry yeast and fermented for 24 h.

## 4. Conclusions

This study explores the effectiveness of a CWK starter culture to create a new sourdough bread. The sensory and the physicochemical properties of sourdough bread formulated using CWK varying in type of starter culture and concentration were determined using the CATA method. The results of CATA analysis showed that breads B (8.30 log CFU/mL of *L. fermentum*), D (4.90 log CFU/mL of *L. fermentum*) and F (9.60 log CFU/mL of *L. plantarum*) were associated with positive sensory attributes. These samples were subjected to further physicochemical characterization. One of the more significant findings of this study was that the bread sample formulated using a CWK culture containing a high concentration of *L. plantarum*, without dry yeast and fermented for 24 h (bread sample F), had high amounts of carboxylic and amino acids, which indicated that *L. plantarum* was an efficient microorganism in terms of carrying out both proteolysis and acidification in the medium during fermentation. This bread was also strongly correlated to most of the positive sensory attributes. The current data highlight the use of CWK starter culture to produce fermented sourdough bread samples, which were further analyzed using the CATA method to profile their sensory properties. As a result of this investigation, suggestions for future research include the use of generic descriptive analyses to provide a better understanding of changes in the intensity of sensory attributes. It is also recommended that future studies include a reference sourdough for direct comparison with samples varying in cultures and inoculum concentrations when carrying out textural analyses. Finally, further work is required to further understand the functional aspects of this CWK fermented sourdough bread that can positively influence the health of consumers.

## Figures and Tables

**Figure 1 foods-09-01165-f001:**
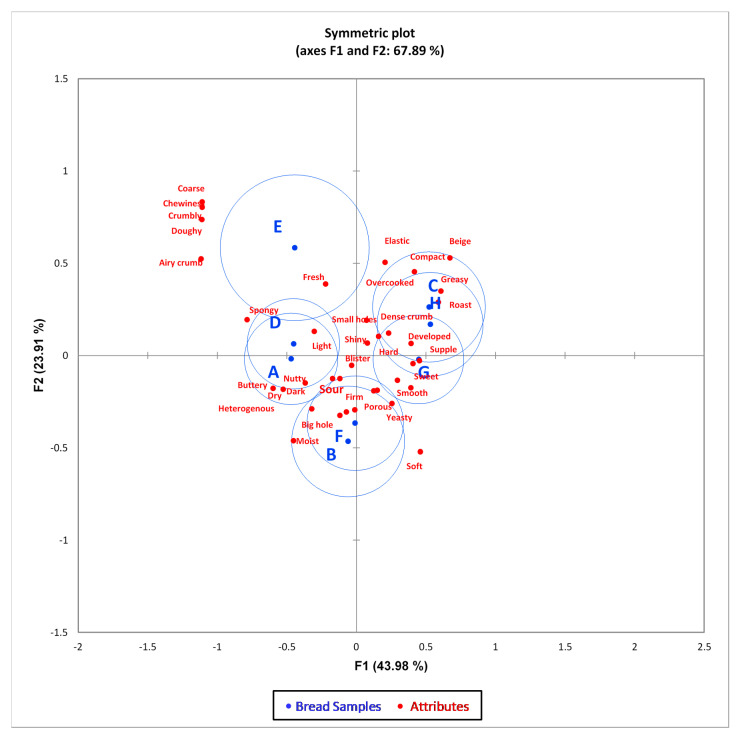
Correspondence analysis plot (with CI 95% based on chi-square distance) of CWK fermented sourdough bread samples A–H, using check-all-that-apply results.

**Figure 2 foods-09-01165-f002:**
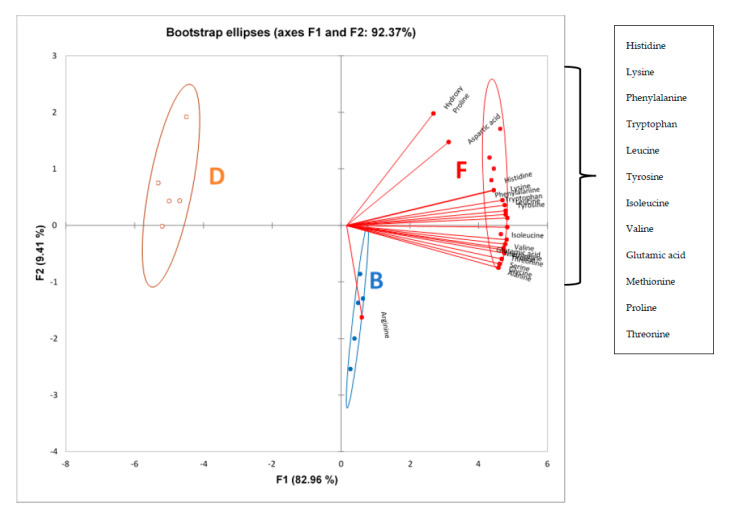
Principal component analysis biplot with bootstrapped confidence ellipses at 95% confidence level for amino acid composition determined using LCMS for sourdough samples B, F and D.

**Table 1 foods-09-01165-t001:** Formulation of sourdough bread using varying concentrations of isolates, yeast and fermentation time using the D-optimal design.

Breads	Flour (g)	Coconut Water Kefir with 12.00 G/L of Added Sucrose (mL)	Sourdough (LAB) (Log CFU/mL)		Salt (g)	Dry Yeast (*Saccharomyces Cerevisiae*) (g)	Time of Fermentation (h)
			Isolate 1 *Lactobacillus fermentum*	Isolate 2 *Lactobacillus plantarum*			
Bread A	500.00	300.00	8.30		8.18	1.16	18
Bread B	500.00	300.00	8.30		8.18	0	24
Bread C	500.00	300.00	4.90		8.18	1.16	18
Bread D	500.00	300.00	4.90		8.18	0	24
Bread E	500.00	300.00		9.60	8.18	1.16	18
Bread F	500.00	300.00		9.60	8.18	0	24
Bread G	500.00	300.00		5.00	8.18	1.16	18
Bread H	500.00	300.00		5.00	8.18	0	24

**Table 2 foods-09-01165-t002:** List and description of sensory attributes used in sensory testing using CATA [39,40].

Attributes	Food Reference	Definition	References
FLAVOUR (Taste and smell)			
Buttery	Mainland unsalted butter (Countdown, NZ)	The aromatics commonly associated with natural, fresh, unsalted butter	[41]
Nutty	Coconut meat–dry (Countdown, NZ)	Taste associated with nuts, like coconut	[42]
Sour	0.1% citric acid solution	Sour taste, like that elicited by citrus fruits	[41]
Coconut	Concentrated coconut water (UFC, NZ)	Coconut water taste	[41]
Roasted	Roasted bread (White bread, Countdown, NZ)	Aroma associated with roasted bread	[42]
Sweet	3% sucrose solution	Sweet taste, like that elicited by sugar	[41]
Salty	0.2% sodium chloride solution	Salty or umami taste–elicited by glutamic acid or NaCl solution	
Floury	Typical of wholemeal flour of wheat mixed with boiling water in proportion 1:2	Aromatics associated with standard baking flour (example: wheat flour)	
Yeasty	A teaspoon of yeast in 250 mL of water	Odor associated with yeast fermentation in bread	
Toasted	Toasted white bread, (Countdown NZ)	The odor impression of bread and crumb after baking/heating	[42]
Fresh		Freshly or recently baked bread loaf	[43]
Fruity	Mixed fruit platter from (Countdown, NZ)	An aromatic impression of dark fruit that may include sweet, slightly brown, overripe or somewhat sour	[42]
APPEARANCE			
Developed	Loaf of fresh bread from Deli (Countdown, NZ)	When the bread is leavened properly and develops small air pockets/small holes	[41]
Overcooked	Overcooked bread	When the bread is cooked beyond the normal time and tastes a bit burnt	[43]
Homogenous		Homogeneity of the pores in the crumb	
Heterogenous		Unevenly distributed pores in the crumb	
Golden		When the color of the crust is light brown/golden upon baking	
Flat		Due to less aeration and leavening of the dough	
Dark		Degree of color darkness in the crumb ranging from white to dark brown	
Thin		Width of the crust is less	
Thick		Width of the crust is bigger	
Smooth		Smoothness/evenness of the crust	
Shiny	Croissant (Countdown, NZ)	Reflection of light on the piece	
Greasy		The appearance of bread which is covered with oil/animal fat later	
Cracked		The split crust due to increased leavening	
Brown/Beige		The color of the bread after baking	
Blister		Formation of tiny bubbles on the bread crust after baking	
Porous		Air pockets in the leavened bread	
Sticky		Bread particles stick to the pallet during mastication and take a longer time to be washed down by the saliva	
Small holes		Pores in the leavened bread	
Dry		Sensation of dryness due to lack of saliva; absence of water	[44]
Compact/Dense crumb		Bread containing little cells filled with gas; high density	[44]
Airy crumb/Big hole		Bread containing cells filled with gas; low density	[44]
TEXTURE			
Supple		Degree to which particles reside on the palate during consumption	[41]
Spongy	Sponge cake (Countdown, NZ)	Like a sponge, especially in being porous, compressible or absorbent.	[44]
Soft	Sandwich bread (Countdown, NZ)	Less force required to bite through sample	
Crunchy		Low pitch sound produced on crust fracture during mastication	
Moist		Amount of moisture perceived on the surface of the product when in contact with the oral cavity	[41]
Hard		Force required to bite completely through sample placed between the molars	
Dry		Bread baked due to loss of moisture	[43]
Doughy		Underbaked bread dough	
Chewiness		Longer time required to chew the bread to reduce it to a consistency suitable for swallowing	
Crumbly		Bread particles fall apart easily	[44]

**Table 3 foods-09-01165-t003:** CATA data by citation frequency and Cochran’s Q test associated with each sample of CWK fermented sourdough bread (sample A–H) by consumers (*n* = 50) on check-all-that-apply questions. ^a–i^: Different superscript letters in rows denote frequencies of attributes that are significantly different across the sourdough bread samples. *p*-value greater than α = 5% indicates no significant difference.

Products/Dimensions	*p*-Values	A	B	C	D	E	F	G	H
**Appearance**									
Airy crumb	0.000	13 ^b^	0 ^a^	0 ^a^	0 ^a^	10 ^b^	10 ^b^	0 ^a^	0 ^a^
Beige	0.000	5 ^b^	0 ^a^	13 ^c^	0 ^a^	12 ^c^	3 ^b^	23 ^d^	24 ^d^
Big hole	0.001	15 ^b,c^	22 ^c^	10 ^b^	13 ^b,c^	4 ^a^	13 ^b,c^	11 ^b^	6 ^a^
Blister	0.000	23 ^e^	22 ^e^	0 ^a^	12 ^d^	8 ^c^	22 ^e^	7 ^c^	1 ^a,b^
Dark	0.000	27 ^e^	17 ^c,d^	7 ^b,c^	20 ^d^	3 ^a^	6 ^b^	15 ^c^	12 ^c^
Dense crumb	0.034	8 ^b^	15 ^d,e^	17 ^e^	14 ^c,d^	8 ^a,b^	5 ^a^	12 ^b,c^	12 ^b,c^
Developed	0.018	4 ^b,c^	5 ^c^	4 ^b,c^	5 ^a,b^	2 ^a^	3 ^a,b^	9 ^d^	12 ^e^
Doughy	0.000	12 ^d^	0 ^a^	0 ^a^	9 ^c^	7 ^b^	0 ^a^	0 ^a^	0 ^a^
Greasy	0.000	3 ^a^	5 ^b,c^	19^c^	3 ^a^	8 ^d^	6 ^c^	14 ^e^	18 ^f^
Heterogenous	0.000	19^g^	16 ^e^	1 ^a^	17 ^e,f^	7 ^d^	7 ^d^	3 ^b,c^	5 ^c^
Light	0.001	12 ^e^	16 ^g^	9 ^a,b,c^	15 ^f,g^	4 ^b^	2 ^a^	9 ^c,d^	10 ^d^
Moist	0.000	9 ^d^	20 ^g^	3 ^a^	6 ^c^	9 ^d^	13 ^e,f^	11 ^d,e^	4 ^b^
Overcooked	0.001	7 ^a^	11 ^c^	8 ^b^	9 ^b,c^	22 ^f^	18 ^e^	17 ^d,e^	18 ^e^
Porous	0.000	16 ^i^	6 ^c,d^	3 ^b,c^	5 ^c^	1 ^a^	13 ^f,g^	12 ^e,f^	14 ^g,h^
Shiny	0.029	15 ^f,g^	7 ^a,b^	14 ^a,b,c^	6 ^a^	12 ^d,e^	19 ^h^	14 ^e,f^	10 ^c,d^
Small holes	0.002	13 ^e,f^	10 ^c,d^	4 ^b^	11 ^d,e^	1 ^a^	15 ^f^	10 ^c,d^	10 ^c,d^
Smooth	0.002	10 ^d,e^	6 ^b^	9 ^c,d^	8 ^b,c^	1 ^a^	15 ^f,g^	16 ^g^	13 ^f^
Supple	0.000	4 ^b,c^	3 ^a,b^	8 ^d^	5 ^c^	2 ^a^	13 ^f,g^	11 ^e^	14 ^g^
Thick	0.067	15	9	10	4	15	14	13	11
Flat	0.068	3	5	8	7	4	10	8	1
Golden	0.103	15	18	15	16	12	14	7	8
Crunch	0.110	5	7	6	13	10	13	12	14
Sticky	0.130	14	9	5	7	14	12	13	11
Cracked	0.221	9	11	6	11	8	5	5	4
Thin	0.244	9	6	9	5	5	1	10	6
Grey	0.523	11	14	7	14	9	11	11	15
Homogenous	0.618	11	16	10	11	10	11	14	15
Brown	0.772	9	11	13	8	7	12	11	11
**Flavor (Taste and smell)**									
Buttery	0.000	15 ^b,c^	10 ^b^	0 ^a^	16 ^b,c^	13 ^b^	22 ^c^	2 ^b^	1 ^a,b^
Nutty	0.000	17 ^c,d^	21 ^f^	18 ^d,e^	19 ^e^	7 ^a^	14 ^b^	16 ^b,c^	7 ^a^
Sour	0.000	5 ^b,c^	12 ^d,e^	11 ^d^	6 ^c^	3 ^a^	17 ^g^	15 ^f^	16 ^f,g^
Roast	0.000	2 ^a^	4 ^b,c^	17 ^f^	4 ^b,c^	6 ^c^	6 ^c^	15 ^e,f^	11 ^d^
Sweet	0.044	5 ^b,c^	5 ^b,c^	9 ^d^	6 ^c^	1 ^a^	9 ^d^	11 ^e^	11 ^e^
Yeasty	0.002	10 ^d^	5 ^b^	4 ^a,b^	5 ^b^	3 ^a^	15 ^f^	13 ^e^	9 ^c,d^
Fresh	0.050	6 ^c,d^	5 ^b,c^	8 ^d^	12 ^e,f^	11 ^e^	3 ^a^	15 ^f^	4 ^a,b^
Fruity	0.057	14	7	6	9	9	6	3	5
Toasted	0.085	11	6	11	6	15	16	9	13
Coconut	0.126	11	13	18	6	9	10	11	10
**Texture**									
Chewiness	0.000	10 ^b^	0 ^a^	0 ^a^	12 ^b,c^	14 ^c^	0 ^a^	0 ^a^	0 ^a^
Coarse	0.000	9 ^b^	0 ^a^	0 ^a^	10 ^b,c^	13 ^c^	0 ^a^	0 ^a^	0 ^a^
Compact	0.002	0 ^a^	3 ^a,b,c^	8 ^b,c,d^	2 ^a,b^	9 ^c,d^	3 ^a,b,c^	11 ^d^	6 ^b,c,d^
Crumbly	0.006	4 ^b,c^	0 ^a^	0 ^a^	3 ^b^	4 ^b,c^	0 ^a^	0 ^a^	0 ^a^
Dry	0.000	13 ^c^	19 ^e^	4 ^a^	17 ^d,e^	7 ^b,c^	15 ^c,d^	4 ^a^	6 ^b^
Elastic	0.000	13 ^e,f^	0 ^a^	15 ^f^	9 ^d^	10 ^d,e^	5 ^c^	4 ^b,c^	22 ^g^
Firm	0.004	9 ^c^	15 ^f,g^	9 ^c^	3 ^a^	4 ^a,b^	16 ^g^	13 ^e^	12 ^d,e^
Hard	0.005	11 ^d,e^	14 ^f,g^	22 ^h^	10 ^c,d^	8 ^a^	9 ^a,b,c^	8 ^a^	16 ^g^
Soft	0.000	1 ^b^	14 ^e,f^	4 ^c^	2 ^b,c^	0 ^a^	15 ^g^	14 ^e,f^	10 ^d^
Spongy	0.004	21 ^f,g^	11 ^c^	7 ^a^	20 ^f^	16 ^e^	12 ^c,d^	9 ^b^	11 ^c^

The compositions of bread samples A–H are shown below: A: Composed of 8.30 log CFU/mL of *L. fermentum*, 1.16 g of dry yeast and fermented for 18 h. B: Composed of 8.30 log CFU/mL of *L. fermentum*, without dry yeast and fermented for 24 h. C: Composed of 4.90 log CFU/mL of *L. fermentum*, 1.16 g of dry yeast and fermented for 18 h. D: Composed of 4.90 log CFU/mL of *L. fermentum*, without dry yeast and fermented for 24 h. E: Composed of 9.60 log CFU/mL of *L. plantarum*, 1.16 g of dry yeast and fermented for 18 h. F: Composed of 9.60 log CFU/mL of *L. plantarum*, without dry yeast and fermented for 24 h. G: Composed of 5.00 log CFU/mL of *L. plantarum*, 1.16 g of dry yeast and fermented for 18 h. H: Composed of 5.00 log CFU/mL of *L. plantarum*, without dry yeast and fermented for 24 h.

**Table 4 foods-09-01165-t004:** Amino acids content of selected sourdough breads (D, B and F) fermented without yeast for 24 h.

**Bread Samples**	**Histidine** **(µM/L)**	**Hydroxy Proline** **(µM/L)**	**Arginine** **(µM/L)**	**Serine** **(µM/L)**	**Glycine** **(µM/L)**	**Aspartic Acid** **(µM/L)**	**Threonine** **(µM/L)**	**Glutamic Acid** **(µM/L)**	**Alanine** **(µM/L)**
**D**	10.4 ± 0.1 ^c^	8 ± 0.2 ^b^	53.8 ± 0.4 ^a^	298.2 ± 2.4 ^c^	354.1 ± 7.9 ^c^	295.9 ± 45.1 ^b^	88.0 ± 2.2 ^c^	121.4 ± 1.5 ^c^	531.2 ± 13.7 ^c^
**F**	11.8 ± 0.3 ^a^	9.5 ± 0.2 ^a^	54 ± 1.4 ^a^	373.1 ± 1.9 ^a^	413.9 ± 2.1 ^a^	350.2 ± 5 ^a^	184.3 ± 2.4 ^a^	166.6 ± 1.8 ^a^	783.1 ± 6.3 ^a^
**B**	10.8 ± 0.1 ^b^	7.2 ± 0.6 ^c^	54.5 ± 0.9 ^a^	358.6 ± 3.8 ^b^	403.1 ± 1.4 ^b^	297.3 ± 6.1 ^b^	160.2 ± 5.7 ^b^	152.8 ± 1.2 ^b^	755.2 ± 4.5 ^b^
***p*-value**	<0.0001	<0.0001	0.537	<0.0001	<0.0001	0.010	<0.0001	<0.0001	<0.0001
**Bread Samples**	**Proline** **(µM/L)**	**Lysine** **(µM/L)**	**Methionine** **(µM/L)**	**Valine** **(µM/L)**	**Tyrosine** **(µM/L)**	**Isoleucine** **(µM/L)**	**Leucine** **(µM/L)**	**Phenylalanine** **(µM/L)**	**Tryptophan** **(µM/L)**
**D**	251.3 ± 5.1 ^c^	31.3 ± 1.4 ^c^	8.30 ± 0.2 ^c^	5.2 ± 0.1 ^c^	4.2 ± 0.1^c^	3.2 ± 0 ^c^	4.6 ± 0.1 ^c^	2.3 ± 0.1 ^c^	8.30 ± 0.2 ^c^
**F**	333.1 ± 2 ^a^	46.1 ± 1.1 ^a^	12.1 ± 0.4 ^a^	10 ± 0.3 ^a^	6.3 ± 0 ^a^	7.6 ± 0.1 ^a^	8.8 ± 0.1 ^a^	5.3 ± 0.1 ^a^	15.0 ± 0.2 ^a^
**B**	312.7 ± 4.8 ^b^	37.4 ± 1.1 ^b^	11.1 ± 0.1 ^b^	8.4 ± 0.1 ^b^	5.3 ± 0.1 ^b^	5.7 ± 0.1 ^b^	6.6 ± 0.2 ^b^	3.6 ± 0.1 ^b^	11.4 ± 0.2 ^b^
***p*-value**	<0.0001	<0.0001	<0.0001	<0.0001	<0.0001	<0.0001	<0.0001	<0.0001	<0.0001

^a–c^: Different superscript letters in columns signify significant differences (*p* < 0.05). Starter used in sourdough bread samples. F: 9.60 log CFU/mL of *L. plantarum*, without dry yeast and fermented for 24 h; B: 8.30 log CFU/mL of *L. fermentum*, without dry yeast and fermented for 24 h; D: 4.90 log CFU/mL of *L. fermentum*, without dry yeast and fermented for 24 h.

**Table 5 foods-09-01165-t005:** Proximate, texture and carboxylic acid analysis for coconut water kefir fermented sourdough bread samples.

Test Name	Bread Sample B	Bread Sample D	Bread Sample F	*p*-Value
Ash Content (g/100 g)	1.80 ± 0.10 ^a^	1.90 ± 0.10 ^a^	1.90 ± 0.20 ^a^	0.422
Carbohydrates (g/100 g)	51.20 ± 0.20 ^a^	53.10 ± 0.40 ^a^	51.90 ± 0.30 ^a^	0.455
Dietary fiber (g/100 g)	3.10 ± 0.10 ^a^	2.90 ± 0.10 ^b^	3.20 ± 0.20 ^a^	0.064
Fat Content (g/100 g)	1.20 ± 0.00 ^a^	1.20 ± 0.00 ^a^	1.40 ± 0.00 ^a^	0.079
Moisture (g/100 g)	33.60 ± 0.10 ^b^	32.60 ± 0.10 ^c^	34.30± 0.10 ^a^	0.551
Protein Content (g/100 g)	8.30 ± 0.30 ^a,b^	8.50 ± 0.50 ^a^	8.20 ± 0.40 ^b^	0.047
Titratable acidity (g/100 g)	0.4 ± 0.00 ^b^	0.40 ± 0.10 ^b^	0.60 ± 0.10 ^a^	0.011
Enumeration of Yeasts and Molds (CFU/g)	<10	<10	<10	
Hardness	937.30 ± 26.40 ^b^	966.30 ± 20.20 ^b^	1137.10 ± 18.30 ^a^	<0.0001
Resilience	37.80 ± 0.60 ^b^	34.90 ± 1.40 ^b^	44.80 ± 3.50 ^a^	0.004
Springiness	76.70 ± 6.30 ^b^	87.20 ± 2.40 ^a^	87.90 ± 1.90 ^a^	0.027
Pyruvic acid (mg/L)	214.67 ± 14.12 ^b^	130.50 ± 3.85 ^c^	242.94 ± 13.99 ^a^	<0.0001
Acetic acid (mg/L)	53.11 ± 2.38 ^b^	42.78 ± 4.07 ^c^	63.39 ± 1.69 ^a^	<0.0001
Succinic acid (mg/L)	45.18 ± 1.16 ^b^	33.88 ± 2.47 ^c^	52.90 ± 2.63 ^a^	<0.0001
Lactic acid (mg/L)	9.27 ± 1.12 ^b^	459.2 ± 11.99 ^c^	743.54 ± 5.80 ^a^	<0.0001

Means in the rows for each determined parameter followed by different letters (^a, b, c^) are significantly different (*p* < 0.05), as determined by the ANOVA. The compositions of samples B, D and F are as shown in the footnote. Starter culture used in bread samples are as follows. F: 9.60 log CFU/mL of *L. plantarum*, without dry yeast and fermented for 24 h; B: 8.30 log CFU/mL of *L. fermentum*, without dry yeast and fermented for 24 h; D: 4.90 log CFU/mL of *L. fermentum*, without dry yeast and fermented for 24 h.

**Table 6 foods-09-01165-t006:** Cosine values between bread product and the attributes used to describe the samples, obtained by Correspondence Analysis (CA) for eight sourdough bread samples using CATA questions.

Attributes/Bread Samples	A	B	C	D	E	F	G	H
Flavor (Taste and smell)								
Buttery	0.969	0.405	−0.984	0.91	0.353	0.308	−0.944	−1
Nutty	0.715	0.808	−0.941	0.58	−0.163	0.743	−0.652	−0.875
Fresh	0.465	−0.796	−0.056	0.615	0.992	−0.854	−0.54	−0.213
Yeasty	−0.59	0.698	0.202	−0.724	−1	0.768	0.658	0.354
Sour	−0.896	0.289	0.631	−0.96	−0.877	0.386	0.931	0.745
Roast	−0.914	−0.551	1	−0.828	−0.192	−0.462	0.876	0.989
Sweet	−0.989	−0.019	0.839	−1	−0.687	0.084	0.998	0.914
Appearance								
Airy crumb	0.961	−0.114	−0.759	0.995	0.778	−0.216	−0.981	−0.852
Heterogenous	0.957	0.445	−0.991	0.89	0.31	0.351	−0.928	−1
Dry	0.942	0.488	−0.996	0.867	0.264	0.396	−0.909	−0.997
Doughy	0.888	−0.306	−0.617	0.955	0.886	−0.403	−0.925	−0.733
Dark	0.828	0.69	−0.986	0.716	0.015	0.612	−0.776	−0.947
Dry	0.767	0.76	−0.964	0.641	−0.086	0.689	−0.708	−0.91
Blister	0.727	0.797	−0.947	0.594	−0.145	0.731	−0.665	−0.884
Light	0.602	0.887	−0.879	0.451	−0.309	0.835	−0.53	−0.793
Moist	0.38	0.975	−0.73	0.21	−0.54	0.947	−0.298	−0.613
Big hole	0.272	0.994	−0.647	0.097	−0.633	0.978	−0.187	−0.518
Small holes	0.087	0.997	−0.492	−0.091	−0.767	1	0.001	−0.349
Overcooked	−0.398	−0.971	0.743	−0.229	0.523	−0.941	0.317	0.628
Porous	−0.518	0.758	0.116	−0.661	−0.998	0.821	0.59	0.272
Compact	−0.703	−0.818	0.935	−0.566	0.18	−0.754	0.638	0.867
Shiny	−0.79	−0.736	0.973	−0.669	0.05	−0.662	0.734	0.925
Beige	−0.809	−0.713	0.98	−0.693	0.017	−0.637	0.755	0.937
Dense crumb	−0.861	−0.644	0.995	−0.758	−0.078	−0.562	0.814	0.966
Greasy	−0.887	−0.603	0.999	−0.791	−0.13	−0.517	0.843	0.978
Smooth	−0.894	0.294	0.627	−0.959	−0.88	0.391	0.929	0.742
Developed	−0.992	−0.288	0.955	−0.954	−0.466	−0.188	0.977	0.99
Supple	−0.995	−0.065	0.863	−0.997	−0.653	0.038	1	0.932
Texture								
Spongy	0.903	−0.275	−0.643	0.965	0.87	−0.372	−0.937	−0.756
Crumbly	0.811	−0.443	−0.495	0.902	0.944	−0.532	−0.859	−0.626
Chewiness	0.786	−0.48	−0.458	0.883	0.957	−0.568	−0.836	−0.593
Hard	−0.904	−0.572	1	−0.813	−0.167	−0.485	0.863	0.985
Soft	−0.63	0.661	0.251	−0.758	−0.997	0.735	0.695	0.401
Elastic	−0.414	−0.966	0.754	−0.246	0.509	−0.935	0.333	0.641
Firm	−0.671	0.62	0.302	−0.791	−0.992	0.698	0.733	0.449
Coarse	0.775	−0.495	−0.443	0.875	0.962	−0.582	−0.827	−0.579

^a^ Values in green indicate high positive correlation of the sensory attribute with the respective sample. ^b^ Values in red indicate high negative correlation of sensory attribute with the respective sample.

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
