# Peer review of "Sensory and Physicochemical Characterization of Sourdough Bread Prepared with a Coconut Water Kefir Starter"

_foods, 2020, doi:10.3390/foods9091165_

Round 1

Reviewer 1 Report

There is no comparison with a traditional sourdough (without CKW) or with CKW (without LAB inoculum) or with LAB inoculum but without CKW. In my opinion only with these combinations is it possible to evaluate the effects (positive or negative) on the addition of CKW and LAB to bread. As organized, the experimental plan does not allow any useful conclusion to be made in order to choose to use the CKW and selected LAB.

The results, supported by an adequate statistical analysis, lack scientific importance

One important parameters, the pH during fermentation, has not been analysed, this parameter could justify many of the results obtained

How has been standardized the CKW before addition of LAB?

How many test (replication) has been done? How many time passed from bread preparation to tasting and analysis? It’s the same for all breads?

the authors do not justify well the results and correlations obtained

in my opinion there are too many descriptors, many of which are useless because they are not directly connected with the factors of the experimental plan (LAB concentration, LAB used, Fermentation time)

Line Item Comments

121- The ingredients are reported as baker’s percentage relative to the amount flour used in the recipe and not the water

200- the CFU / mL values ​​are correct?

436 – The sentence is correct, so why this difference?

514 – the sentence it’s not clear. “L. plantarum had a Higher acidification rate than the chemically acidified sourdoughs”

582 – Marco Gobetti

Author Response

Response to Reviewer 1 Comments

  • There is no comparison with a traditional sourdough (without CKW) or with CKW

(without LAB inoculum) or with LAB inoculum but without CKW. In my opinion only

with these combinations is it possible to evaluate the effects (positive or negative) on the addition of CKW and LAB to bread. As organized, the experimental plan does not allow any useful conclusion to be made in order to choose to use the CKW and selected LAB.

Response 1. We would like to thank the reviewer for the comments. With regards to the first point on comparison, pilot studies were carried out for traditional sourdough (without CWK) and with CWK (without LAB inoculum), however, the results did not relate to the goal of this study which was to use CWK based starter and LAB isolates to prepare a sourdough and investigate its sensory and physicochemical properties. Therefore, the results from the pilot study are not included in this manuscript.

  • The results, supported by an adequate statistical analysis, lack scientific importance.

One important parameters, the pH during fermentation, has not been analysed, this

parameter could justify many of the results obtained.

Response 2. The pH had been analysed in this study, but the results of which were not significant for samples B, D and F, as shown below. All the samples which were fermented for 18 h had a significantly high pH compared to all the samples fermented for 24 h.

Would you like me to add these results to the manuscript?

Bread sample

Time of fermentation (h)

pH

A

18

4.1a

B

24

3.9a,b

C

18

4.1a

D

24

3.9a,b

E

18

4.1a

F

24

3.8b

G

18

4.1a

H

24

3.8b

P-value

0.008

a, b: Different superscript letters in column refers to the significantly different pH across the sourdough bread samples fermented at 18 and 24h. P-value greater than α = 0.05 indicates no significant difference.

  • How has been standardized the CKW before addition of LAB?

Response 3. Commercially available coconut water, kefir grains and sucrose were used to prepare CWK, which is why this combination was standardised. Changes have been made in L129-L132 - “CWK was prepared using 300 ml of UFC© Coconut water (manufactured by Universal Food Public Company Limited, Thailand, purchased at Countdown, Auckland City) and 1.5 g/L of kefir starter (Body Ecology™, New Zealand), and 12g/L of sucrose (Food grade, Chelsea sugar limited, New Zealand).”

  • How many test (replication) has been done? How many time passed from bread preparation to tasting and analysis? It’s the same for all breads.

Response 4.Three test replicates had been done – this has been included in the materials and method’s section as below:

L144-145 and L 196-197 – “Three experimental replicates for each formulation were tested.”

For time between the bread preparation to tasting, the breads were analysed within 12 hours from baking:

L166-L167 – “Once all the bread samples were prepared, they were analysed using CATA method within 12 hours of baking, which was the standard across all bread samples.”

  • The authors do not justify well the results and correlations obtained. In my opinion there are too many descriptors, many of which are useless because they are not directly connected with the factors of the experimental plan (LAB concentration, LAB used, Fermentation time).

Response 5. We agree with the reviewer, and therefore, to make it easy for the reader, each descriptor has been categorised into modalities such as flavour, appearance, and texture. The non-significant values have been shifted to the bottom of each category. Table 2, 3 and 4 have been updated, and reflect all the changes.

Also, the concentration and the type of LAB used has an impact on the on the carboxylic acids produced in the sourdough, which is explained as below. Fermentation time did not have much impact on the sourdough, since this parameter was constant (24 h for bread samples B, D and F). IT has been discussed that there is higher concentration of carboxylic acids produced in the bread sample formulated with high concentration of LAB, specifically, L. Plantarum, as below –

L596-L603 – “According to Table 5 sample F had the significantly highest concentration of lactic acid, acetic acid, succinic acid and pyruvic acid, followed by sample B, and Sample D. Sample F had a high concentration (9.6 log CFU/ml) of L. plantarum cells and sample B had a high concentration (8.3 log CFU/ml) of L. fermentum. The high carboxylic acids content in samples F and B are attributed to the high concentrations of LAB isolates. Hadaegh et al., (2017) [41] reported increased production in lactic and acetic acids in the sourdoughs inoculated with high concentration (30%) of starter cultures like L. plantarum jQ 301799, L. casei jQ412732 and/or L. brevis IBRC-M10790, compared to sourdoughs inoculated with 20% inoculum.”

Additionally, wherever applicable, correlations between the CATA results and concentration and type of LAB have also been made.

  • 121- The ingredients are reported as baker’s percentage relative to the amount flour used in the recipe and not the water

Response 6.The concentration of baker’s yeast relative to the amount of flour and CWK added to prepare the sourdough, which was 1.6 g. The following lines have been changed from yeast =1.6 g/L to only 1.6 g. Lines reflecting these changes are: L115, L124

  • 200- the CFU / mL values are correct?

Response 7.Three corrections have been made that now reflect the correct concentration of the cfu/ml values-

L201-L205 - “The sourdough samples B, D and F were used for this analysis. The reason for this selection is stated in section 3.1. Sample F was made from sourdough containing 9.6 CFU/ml of L. plantarum, without dry yeast and was fermented for 24 h. Samples B and D were made from sourdough containing 8.3 CFU/ml and 4.9 CFU/ml of L. fermentum respectively, without dry yeast and fermented for 24 h.”

  • 436 – The sentence is correct, so why this difference?

Response 8. The bread samples that are heterogenous could be explained by the variation in the overall crust colour of the baked bread, which can be explained by the very small and insignificant changes in the temperature of the baking the breads or the change in the batch numbers of the bread.

‘Light’ is an attribute which defines the appearance of the bread after it is baked.

Explanation has been provided for the L436 – which is now L459

L459 -L463 - “ The heterogenous and light appearance of the bread are dependent on the baking temperature of bread, which could be due to the flavour compounds formed during the Maillard reactions such as furans, pyrroles (thiophenes, thiophenones, thiapyrans and thiazolines), pyrazines, sulphur containing compounds, and, lipid degradation products such as alkanals, 2-alkenals and 2,4-alkadienals [11].”

Please advise if you would recommend including more explanation in the text.

  • 514 – the sentence it’s not clear. “L. plantarum had a Higher acidification rate than the chemically acidified sourdoughs”

Response 9. L514, which is now line 550, has been amended to:

L 549-L550 – “Bello et al., (2007) [29] have reported that the sourdough bread fermented with L. plantarum had a higher acidification rate compared to the chemically acidified sourdoughs which are acidified using acids such as citric acids.”

  • 582 – Marco Gobetti

Response 10. The change to now L621 has been made by deleting ‘Marco’ from the above reference. L621 – “A study carried out by Gobbetti et al. (1994) [39] have reported”

Reviewer 2 Report

l.12 – please list at least three products ‘… from various sources such as kefir’

l.18 – please rewrite ‘… were further chosen for further physicochemical analysis’

  1. 20-21 – ‘… were carried out on breads B, D and F…’ unfortunately, no one knows what is the bread B, D and F – please remove ‘B, D and F’
  2. 111-113 – please explain the reason of difference in time fermentation for breads with and without yeast addition

Table 1 – please unify description of breads – ‘Bread 1’, Bread A’ or Bread 1 A’ – choose one of them

Table 3 – there are 84 rows. Why? It seems that some repetitions are present. If there are no significant differences for p > than 0,05, what is the reason to present suprerscipts (e.g. sweet)

Major comments:

  1. Please rewrite abstract replacing terms bread B, D, and F by more understandable description. Also it would be worth to point out what improvements or achievements have been observed. The term (l.29) ‘… may improve …‘ without any other justification gives impression that authors are not sure if their observation are worth to publish. Please add in a simple way the main finding of your research
  2. The analysis provided for sensory analysis is hard to be understand for the reader. The table 3 consist of some repetitions and it is not possible to follow results used to remove selected samples from further analysis. Additionally, there is a need to get some explanations regarding applying CATA. Based on table 3, It seems that there is no feature with more than 46% of votes. Authors should extend the scope of research by applying the intensity of the taste, small or color as well or justify the results obtained (wrong panelists training programme?). Also results that the same sample is simultaneously heterogenous and homogenous at the same level (sample B) needs some explanation – especially because only 32% of panelists pointed out such feature (what about other 36%). No choice means that panelist was not properly trained. Numbers placed in the table 3 have to be rechecked and statistics should be recalculated (e.g. results for ‘buttery’ feature from 0 to 1 no significant difference, and from 2 to 16 no significant difference(?); e.g. for ‘hard’ feature, sample E (value 8 marked by letter a), for sample G (value 8 (the same as for E), marked by letter b,c – is that true that there is a significant difference between 8 and 8? Such situation can be found very often in the table. Unfortunately, because such results had impact into the further analysis there is a need to recheck results and statistics and repeat analysis for the second stage if there is a need to do so. It would be worth for a potential reader to add the final table with desired sensory properties of samples (from CATA) chosen for physicochemical analysis with the percentage value of consumer acceptability.
  3. Please rewrite conclusions to point out for a potential reader if the goal of the research has been achieved and what are the main findings. The goal was to ‘investigate the use a coconut water kefir-based fermentation starter culture using Lactobacillus fermentum and Lactobacillus plantarum to develop a sourdough bread. Can you explain the statement (l. 621-623) ‘success … to produce … product with desirable sensory and physicochemical properties’ From the analysis presented in the paper it is not possible to find justification of that. It is hard to find the sample with ‘desirable sensory and physicochemical properties’. Please rewrite.

Author Response

Response to Reviewer 2 Comments

Reviewer #2

  • 12 – please list at least three products ‘… from various sources such as kefir’

Response 1. We would like to thank the reviewer for their comments.

The addition has been made to L11 -L12 – “There is a recognized need for formulating functional food products using selected Lactic acid bacteria (LAB) starter cultures from various sources such as kefir, yoghurt or kombucha.”

  • 18 – please rewrite ‘… were further chosen for further physicochemical analysis’

Response 2. For L18-L19, the repetitive word ‘further’ has been deleted – “three bread samples with positive sensory attributes were further chosen for further physicochemical analysis chosen for physicochemical analysis”

  • 20-21 – ‘… were carried out on breads B, D and F…’ unfortunately, no one knows what is the bread B, D and F – please remove ‘B, D and F’

Response 3. We have removed the sample names B, D and F from L 20-21.

The corrections have been applied throughout L19-L31 – “Physicochemical analysis (texture, proximate composition, shelf life, carboxylic acid analysis and amino acid analysis) were carried out on breads formulated with starter culture concentrations of 8.3 log CFU/ml of L. fermentum, 4.9 log CFU/ml of L. fermentum, and 9.6 log CFU/ml of L. plantarum, each fermented for 24 h without baker’s yeast. Bread sample that was formulated with a coconut water kefir (CWK) starter culture containing 9.6 log CFU/ml of L. plantarum, without dry yeast and fermented for 24h had significantly high values for almost all amino acids and low protein content compared to bread samples formulated using CWK cultures containing 8.3 log CFU/ml of L. fermentum and 4.9 log CFU/ml of L. fermentum, both without dry yeast and fermented for 24h. Bread sample formulated with CWK starter culture containing 9.6 log CFU/ml of L. plantarum, without dry yeast and fermented for 24h F also produced significantly high quantities of organic acids (pyruvic acid, acetic acid, lactic acid and succinic acid. These changes in the physicochemical properties can improve the overall bread quality in terms of flavour, shelf life, texture and nutritional value.”

  • 111-113 – please explain the reason of difference in time fermentation for breads with and without yeast addition.

Response 4. Thank you for the comment, we agree with the reviewer and have therefore made the corrections in the lines below:

L125-L128 – “Different fermentation times of 18h and 24 h were used for fermenting the sourdough bread samples to determine the best condition to achieve high acidity, low pH and high dough volume. Baker’s yeast was only added to samples fermented for 18 h for the purpose of speeding up fermentation.”

  • Table 1 – please unify description of breads – ‘Bread 1’, Bread A’ or Bread 1 A’ – choose one of them

Response 5. Table 1 has now been updated and the eight bread samples are referred to as A, B, C … G, H in the table.

  • Table 3 – there are 84 rows. Why? It seems that some repetitions are present. If there are no significant differences for p > than 0,05, what is the reason to present superscripts (e.g. sweet).

Response 6. We sincerely apologise for the repetitions in Table 3. These repetitions have now been rectified and removed from the table. The table has now been separated in terms of modalities (Appearance, Taste, and Texture) and all the nonsignificant attributes have been moved to the bottom of the table for each of the modalities and the superscripts have been removed.

  • Please rewrite abstract replacing terms bread B, D, and F by more understandable description. Also, it would be worth to point out what improvements or achievements have been observed. The term (l.29) ‘… may improve …‘ without any other justification gives impression that authors are not sure if their observation are worth to publish. Please add in a simple way the main finding of your research

Response 7. The abstract has been re-written using a more understandable descriptions for bread samples B, D and F. the main findings have been presented in a simpler way, as below:

L11-L31 –

“There is a recognized need for formulating functional food products using selected Lactic acid bacteria (LAB) starter cultures from various sources such as kefir, yoghurt or kombucha. The principle objective of this study was to investigate the use of a coconut water kefir-based fermentation starter culture using Lactobacillus fermentum and Lactobacillus plantarum to develop a sourdough bread. Check-all-that-apply (CATA) sensory profiling was used to evaluate the sensory profile of sourdough breads that varied with culture type, culture concentrations, with and without added yeast, and fermentation for 18 and 24 h. Based on correspondence analysis (CA) of the CATA results, three bread samples with positive sensory attributes were further chosen for physicochemical analysis. Physicochemical analysis (texture, proximate composition, shelf life, carboxylic acid analysis and amino acid analysis) were carried out on breads formulated with starter culture concentrations of 8.3 log CFU/ml of L. fermentum, 4.9 log CFU/ml of L. fermentum, and 9.6 log CFU/ml of L. plantarum, each fermented for 24 h without baker’s yeast.. Bread sample that was formulated with a coconut water kefir (CWK) starter culture containing 9.6 log CFU/ml of L. plantarum, without dry yeast and fermented for 24h had significantly high values for almost all amino acids and low protein content compared to bread samples formulated using CWK cultures containing 8.3 log CFU/ml of L. fermentum and 4.9 log CFU/ml of L. fermentum, both without dry yeast and fermented for 24h. Bread sample formulated with CWK starter culture containing 9.6 log CFU/ml of L. plantarum, without dry yeast and fermented for 24h  also produced significantly high quantities of organic acids (pyruvic acid, acetic acid, lactic acid and succinic acid. These changes in the physicochemical properties can improve the overall bread quality in terms of flavour, shelf life, texture and nutritional value.”

  • The analysis provided for sensory analysis is hard to be understand for the reader.
  • The table 3 consist of some repetitions and it is not possible to follow results used to remove selected samples from further analysis.

Response 8. We agree with the reviewer about the sensory results, therefore, Tables 2, 3 and 4 have been updated to reflect the different modalities. Each attribute has been separated into three main categories – appearance, flavour and texture.

Response 8. For Table 3, we sincerely apologise for copying the table incorrectly. All the repetitions have now been removed and the significant attributes have been placed on the top of the table for each modality, followed by the non-significant attributes. This makes it easier for the reader to follow the results.

  • Additionally, there is a need to get some explanations regarding applying CATA. Based on table 3, It seems that there is no feature with more than 46% of votes.

Response 8. We would like to thank reviewer for the comment, however, would like to explain as below:

CATA provides panellist the ability to check or unchecked an attribute. The panellists were also provided a comprehensive amount of attributes for the bread and may have different views on what describes the sourdough samples or not. The 46% is not the vote, it is a statistical difference between the results obtained by CATA analysis. It means that 46% of the total 50 participants selected the particular attribute for the bread sample being analysed. In CATA method of analysis, the panellist can select as many attributes they like when carrying out the sensory test for the sample in question.

  • Authors should extend the scope of research by applying the intensity of the taste, small or color as well or justify the results obtained (wrong panelists training programme?).

Response 8. Check-All-That-Apply (CATA) analysis was carried out with semi-trained panellists.

CATA is different to Quantitative Descriptive sensory Analysis (QDA), as a questionnaire is given to the panellists in CATA, which has a list of attributes that could be based on appearance, taste (Flavour and smell) and texture. No hedonic scales were used to rate the intensity of each attribute, which are used in QDA. Only a lit of attributes was provided to the semi-trained panellist for CATA.

The panellists can select as many attributes they think apply to the product being tested (Ares et al., 2015; Mello, Almeida, & Melo, 2019). The panellist in CATA can be trained to understand how to select the attributes when carrying out the sensory analysis of the sample. The CATA lists provide multivariate binary data that indicate the applicability of the descriptors provided for the samples.

Furthermore, The Cochran´s Q test was performed to identify significant differences between samples for each attribute included in the CATA analysis.

Ares, G., Antúnez, L., Bruzzone, F., Vidal, L., Giménez, A., Pineau, B., . . . Chheang, S. L. (2015). Comparison of sensory product profiles generated by trained assessors and consumers using CATA questions: Four case studies with complex and/or similar samples. Food quality and preference, 45, 75-86.

Mello, L. S. S., Almeida, E. L., & Melo, L. (2019). Discrimination of sensory attributes by trained assessors and consumers in semi-sweet hard dough biscuits and their drivers of liking and disliking. Food Research International, 122, 599-609. https://doi.org/https://doi.org/10.1016/j.foodres.2019.01.031

  • Also results that the same sample is simultaneously heterogenous and homogenous at the same level (sample B) needs some explanation – especially because only 32% of panelists pointed out such feature (what about other 36%). No choice means that panelist was not properly trained.

Response 8. We would like to thank the review for making this interesting observation. To explain this further, there were a total of 50 participants (semi-trained panellists). Sample B had 16 participants (32%) selecting homogenous and anther 16 (32%) selecting heterogenous – where the other 18 (36%) did not select it.

CATA method consists of presenting to the judges a compilation of attributes of the product subjected to evaluation and they are asked to select those attributes considered appropriate to describe the sample.

It is important to note that the CATA profiling was done per each sensory modality (i.e. appearance), therefore the participants might actually missed the aforementioned attribute as they think that it might not exist/important highlighting the fallback of CATA analysis as to classical descriptive analysis.

To add clarity for the reader on how we carried out the analysis, we have now sub sectioned the sensory results to each sensory modality.

  • Numbers placed in the table 3 have to be rechecked and statistics should be recalculated (e.g. results for ‘buttery’ feature from 0 to 1 no significant difference, and from 2 to 16 no significant difference(?); e.g. for ‘hard’ feature, sample E (value 8 marked by letter a), for sample G (value 8 (the same as for E), marked by letter b,c – is that true that there is a significant difference between 8 and 8? Such situation can be found very often in the table.

Response 8. We sincerely apologise for this mistake, which was made due to the incorrect table being copied to the manuscript. The correct data has now been presented in the table and the has now been corrected. The Table 3 has now been updated with correct statistics.

  • Unfortunately, because such results had impact into the further analysis there is a need to recheck results and statistics and repeat analysis for the second stage if there is a need to do so. It would be worth for a potential reader to add the final table with desired sensory properties of samples (from CATA) chosen for physicochemical analysis with the percentage value of consumer acceptability.

Response 8. We assure the reviewer that the statistical mistakes have been rectified and ensure that it has not caused any impact to the results. CATA method consists of presenting to the judges a compilation of attributes of the product subjected to evaluation and they are asked to select those attributes considered appropriate to describe the sample. The % value in Consumer acceptability is not being studied as no hedonic scale was used to assess the intensity of the attribute. We are only profiling the attributes.

  • Please rewrite conclusions to point out for a potential reader if the goal of the research has been achieved and what are the main findings. The goal was to ‘investigate the use a coconut water kefir-based fermentation starter culture using Lactobacillus fermentum and Lactobacillus plantarum to develop a sourdough bread. Can you explain the statement (l. 621-623) ‘success … to produce … product with desirable sensory and physicochemical properties’ From the analysis presented in the paper it is not possible to find justification of that. It is hard to find the sample with ‘desirable sensory and physicochemical properties’. Please rewrite.

Response 8. The goal of this study was “This study was aimed at developing a sourdough bread produced from a novel sourdough fermented with CWK cultures containing two selected lactic acid bacteria strains (Lactobacillus fermentum and Lactobacillus plantarum) that had high production of phytase and glutamic acid. Sensory analysis using Check-All-That-Apply sensory profiling method was carried out on the baked sourdough breads. Bread samples with desirable sensory properties were then subjected to further physicochemical testing.”

The conclusion has now been amended from L561-L669 - “ This study explores the effectiveness of coconut water kefir starter culture to create a new sourdough bread. Sensory and the physicochemical properties of sourdough bread formulated using coconut water kefir varying in type of starter culture and concentration were determined using the CATA method. Results of CATA analysis showed that breads B (8.3 log CFU/ml of L. fermentum), D (4.9 log CFU/ml of L. fermentum) and F (9.6 log CFU/ml of L. plantarum) had positive sensory attributes. These samples were subjected to further physicochemical characterization. Bread sample F had significantly high values for hardness, whereas bread samples B and D had significantly high values for adhesiveness. One of the more significant findings of this study was that the bread sample formulated using a coconut water kefir culture containing high concentration of L. plantarum, without dry yeast and fermented for 24h had high amounts of carboxylic acids and amino acids, which indicated that L. plantarum was an efficient microorganism which carried out both proteolysis and acidification in the medium during fermentation (bread sample F). Therefore, bread sample F was the sample to which the positive sensory attributes were correlated to, and also had high concentrations of both amino acids and carboxylic acids, and was hard. The current data highlighted the use of CWK starter culture to produce fermented sourdough bread samples, which were further analysed using the CATA method to profile their sensory properties. Further work is required in order to understand the functional aspects of this CWK fermented sourdough bread which may influence health of consumers positively.”

Reviewer 3 Report

The submitted manuscript contains several errors which must be corrected.

The introduction lacks key information as to why coconut water is an area of interest in finding new ingredients for sourdoughing.

There is no information on how kefir was made on coconut water. The authors describe only the isolation of two strains. 

Table 2 and Table 4 should be withdrown from manuscript and located as Supplementary material.  

The metodological error was done as for bread using 6,25 factor for protein recalculation from nitrogen is incorrect. This impacts the protein level calculation and all the statistics done.

The article is written in a superficial way, the authors do not try to explain the causes of the observed phenomena. They only summarize the results and quote other works that received similar results. This is not how a scientific article is written, but a note from the research, or a description of the results. There is no substantive discussion in the area of physicochemical parameters and sensory evaluation. The article is descriptive - there is no reliable discussion of the results.

Translated with www.DeepL.com/Translator (free version)

Line 565 - This is repetition.

Line 531-532 - "The high values for hardness and springiness in sample F could be attributed to high acidity in the bread sample, supported by TTA results described in section 3.2.2." - please provide any explanation for this finding or remove it. 

Author Response

Response to Reviewer 3 Comments

Reviewer #3

  • The introduction lacks key information as to why coconut water is an area of interest in finding new ingredients for sourdoughing.

Response 1. We would like to thank the reviewer for their comments.

Coconut water has been used in this study to serve as a medium for fermentation and growth of kefir grains. The fermented CWK is added to the sourdough. The importance of using coconut water for fermenting the kefir grains has been explained in

L78-79 and L81-83 – “The liquid endosperm of coconut water is of cytoplasmic origin, which fills the cavity within the coconut [47]. The coconut water is of key importance since it contains almost all components of the vitamin B group (except for B6 and B12), minerals, proteins, sugars, amino acids, magnesium, vitamin C, potassium, and growth factors such as auxins, cytokinins and gibberellins, which makes it biologically favourable for human nutrition as well as for the growth of microorganisms, making it an ideal medium for kefir fermentation [7, 54, 64, 80, 61]. This isotonic drink is very low in fat content and also contains optimum amounts of RNA phosphorous which play an active role in the transport of amino acids and respiratory metabolism in living cells [21].”

  • There is no information on how kefir was made on coconut water. The authors describe only the isolation of two strains.

Response 2. Changes have also been made in Materials and Methods section to explain that CWK was added to prepare the sourdough, section 2.2, L129- L137 – “CWK was prepared using 300 ml of UFC© Coconut water (manufactured by Universal Food Public Company Limited, Thailand, purchased at Countdown, Auckland City) and 1.5 g/L of kefir starter (Body Ecology™, New Zealand), and 12g/L of sucrose (Food grade, Chelsea sugar limited, New Zealand). Coconut water kefir was allowed to ferment for up to 48 h at 30°C in a LabServ incubator (Thermo Fisher, New Zealand) before being used in bread preparation. The fermented mixture of coconut water kefir used in this recipe contained 12g/L of sucrose as it produced the highest cell counts for LAB and yeast (data not shown), and faster sugar utilization that resulted in low concentration of residual sugar at the end of 96 h of fermentation, which is why it was used to prepare the sourdough.”

Many more strains had been isolated from the CWK, however, they have not been reported in this manuscript. The reason for selecting only these two strains for this study is because of their ability to produce significant quantities of phytase enzyme and glutamic acid (data not shown – please let us know if you would like us to report the data), both of which are important factors that improve the functional properties of the formulated sourdough. The isolation method has been added in brief in the lines below:

L95-L103 – “Pure cultures of two LAB isolates with ability to produce high concentrations of glutamic acid and phytase enzyme were supplemented at known cell concentrations into the sourdough. The isolates were: L. fermentum and L. plantarum, that were isolated, identified and purified from CWK. The identification was carried out from the fermented CWK using DNA extraction using PowerFood DNA Isolation kit (Mo Bio Laboratories, Carlsbad, CA, USA) and Sanger’s method for DNA sequencing techniques. The raw data (DNA sequences) were received and analysed using Geneious prime Bioinformatics Software Pro 5.6 (Geneious, New Zealand), and identified using the National Center for Biotechnology Information- Basic local alignment search tool (NCBI) BLAST.”

  • Table 2 and Table 4 should be withdrawn from manuscript and located as Supplementary material.

Response 3.We thank the reviewer for this comment, but would like to request the reviewer to reconsider, as both Tables 2 and 4 outline the important attributes that are required to explain this study properly and also for the better flow of the results. Table 2 provides all the definitions for the list of the attributes that were chosen for this study. Whereas, Table 4 shows the cosine values between bread products and the attributes used to describe the samples, obtained by Correspondence Analysis (CA). Cosine values are important as they can fall between −1 and +1. The angle between the vectors (or its cosine) in the full-dimensional space gives the complete information about the relationship between products and attributes.

  • The methodological error was done as for bread using 6.25 factor for protein recalculation from nitrogen is incorrect. This impacts the protein level calculation and all the statistics done.

Response 4. We sincerely apologise to the reviewer for making this mistake in the factor for calculation when writing the method for determination of protein. The factor that was indeed used by the authors was 5.7, but was incorrectly reported in the methods at 6.25. No changes have been made to the results, since the calculations had been made using the factor 5.7 and not 6.25. This change has been made in L258 – “Protein % = (VA-VB) * 1.4007 * M * (5.70/g test portion)”

  • The article is written in a superficial way, the authors do not try to explain the causes of the observed phenomena. They only summarize the results and quote other works that received similar results. This is not how a scientific article is written, but a note from the research, or a description of the results. There is no substantive discussion in the area of physicochemical parameters and sensory evaluation. The article is descriptive - there is no reliable discussion of the results.

Response 5. We would like to thank the reviewer for this constructive comment. We have made additions for the results and discussion part, throughout the manuscript, to explain the phenomenon and have made an attempt to explain all the results with more detail and description. The following lines have been changed, and/or added:

L379-L382 - “Pores are formed due to the microbial activity of the LAB and yeast microorganisms during fermentation of the dough, due to the production of carbon dioxide as the byproduct of homo-fermentation or hetero-fermentation pathways [48, 51, 88, 90].”

L387-L390 – “Maillard reactions determine the browning intensities of the bread surface. It can be observed that bread sample F had significantly high concentrations of the important amino acids, as shown in Table 6, that participate in the Maillard reactions – such as lysine, glycine, tryptophan and tyrosine [11].”

L395-L400 – “Bread samples A, B, C, D, F, G and H were sour compared to bread E. Sour flavour in the breads could be attributed to the addition of lactic acid bacteria, which is brought about by acidification and proteolysis in the sourdough, as a result of fermentation, as shown in Table 5 (Total Titratable Acidity) and Table 6 (Amino acids). Proteolysis during sour-dough fermentation produces amino acids, which are well-known precursors in bread flavour formation [49] and the acidification is the result of the formation of acids such as lactic acid and acetic acid, depending on the strain of LAB and yeast used for fermentation of the dough [49].”

L410-L417 – “Bread samples B and F had significantly firmer texture compared to sample D. This can be explained by the process of retrogradation, in which high concentration of amino acids such as aspartic acid and glutamic acid (Table 6) play an important role in increasing the retrogradation process. Aspartic acid and glutamic acid are both acidic amino acids, which can decrease the swelling power of the flour particles and cause increased gelatinization, syneresis and amylose leaching [20, 88]. Both bread samples F and B have significantly high concentrations of both glutamic acid and aspartic acid in the bread, which could explain the firmer texture of these breads, compared to sample D.”

L436 – L440 – “The impact of sourdough on bread volume has been suggested to be principally due to enzymatic reactions taking place throughout the fermentation. These reactions are aided by the cereal enzymes which influence the metabolism of the LAB and yeast present in the dough during homo or hetero-lactic fermentation, and produce by-products such as carbon dioxide [25, 23, 48].”

L465 – L468 – “The heterogenous and light appearance of the bread are dependent on the baking temperature of bread, which could be due to the flavour compounds formed during the Maillard reactions such as furans, pyrroles (thiophenes, thiophenones, thiapyrans and thiazolines), pyrazines, sulphur containing compounds, and, lipid degradation products such as alkanals, 2-alkenals and 2,4-alkadienals [11].”

L477 – L484 – “Furthermore, bread samples B and F have significantly high concentration of amino acids such as lysine, glycine, tryptophan and arginine, which are involved in the Maillard reactions to produce important flavour compounds which positively contribute to the fresh and nutty attributes [11]. These flavour compounds are either volatile compounds or non-volatile compounds such as 2,3-butanedione, 1-octen-3-one, phenylacetaldehyde, (Z)- 2-nonenal, and 2-ethyl-3-methyl-pyrazine that contribute to the sweet aroma and odour; 2-acetyl-1-pyrroline that contributes to the roasted flavour; 2-octenal that contributes to nutty flavour; and ethyl 2-methylpropanoate that contributes to the fruity flavour in the bread [11].”

L577 – L580 – “Hardness can also be attributed to the process of retrogradation, which is caused by the interaction of starch with amino acids such as aspartic acid and glutamic acid [92]. Sample F had high concentrations of both these amino acids, which could explain the high values for hardness in the sample (Table 6).”

  • Line 565 - This is repetition.

Response 6. L 565, now line 609, has been deleted. “D- Composed of 4.9 log CFU/ml of L. fermentum, without dry yeast and was fermented for 24h.”

  • Line 531-532 - "The high values for hardness and springiness in sample F could be attributed to high acidity in the bread sample, supported by TTA results described in section 3.2.2." - please provide any explanation for this finding or remove it.

Response 7. We thank the reviewer for this comment. Explanation for L531-532’, which is now L572-573, has been amended as below:

L571-L580 – “The high values for hardness and springiness in sample F could be attributed to high acidity in the bread sample, supported by the high values for total titratable acids shown in Table 5.  The higher concentration of L. plantarum F could be responsible for the higher acidification and thus, the higher hardness and springiness value. These results are supported by a study carried out by Bello et al., (2007) [29], which reports a higher acidification rate for sourdough fermented with L. plantarum when compared to the chemically acidified sourdoughs. Hardness can also be attributed to the process of retrogradation, which is caused by the interaction of starch with amino acids such as aspartic acid and glutamic acid [92]. Sample F had high concentrations of both these amino acids, which could explain the high values for hardness in the sample (Table 6).”

Round 2

Reviewer 1 Report

the manuscript has been improved,
but personally I would have preferred fewer parameters for the CATA
analysis, and others such as TPA are not useful
if not compared to a standard sourdough bread.

Author Response

Responses to Reviewer 1 comments:

Point 1: The manuscript has been improved, but personally I would have preferred fewer parameters for the CATA analysis, and others such as TPA are not useful if not compared to a standard sourdough bread.

Answer 1: The authors thank the Reviewer for their comments and agree that fewer parameters could be analysed for CATA method. Methods such as CATA have emerged, in fact, as complementary tools to sensory and consumer science, as they can be applied to gather product descriptions directly from consumers, with the added benefit of having direct feedback from them, and sometimes with their own vocabulary (Moussaoui & Varela, 2010).  Since most of these attributes are used for describing a sourdough bread sample, they were used in this study to provide an exhaustive array of attributes for the consumer panelists to choose from. Hence it is not possible to have fewer sensory attributes. If quantitative descriptive analysis is used however for sensory testing, such a vocabulary can be refined to reduce redundant terms.

With regards to TPA, a recommendation has been made for future research scope to compare it to a standard sourdough bread. This has been added in the ‘Conclusion’ section as shown below:

L682- L688: “As a result of this investigation, suggestion for future research include the analysis of sourdough breads using generic descriptive analysis to provide a better understanding of changes in the intensity of sensory attributes. It is also recommended in future studies to include a reference sourdough for direct comparison with samples varying in cultures and inoculum concentration when carrying textural analysis using the TPA method. Further work is also required to further understand the functional aspects of this CWK fermented sourdough bread, which may influence health of consumers positively.”

References:

Moussaoui, K. A., & Varela, P. (2010). Exploring consumer product profiling techniques and their linkage to a quantitative descriptive analysis. Food Quality and Preference, 21(8), 1088-1099.

Crowley, P., Schober, T. J., Clarke, C. I., & Arendt, E. K. (2002). The effect of storage time on textural and crumb grain characteristics of sourdough wheat bread. European Food Research and Technology, 214(6), 489-496.

Reviewer 2 Report

Dear editor,

After modifications introduced by authors, the manuscript seems to be more readable. Results are better justified and described, but besides of that there is a suggestion to present results and findings in more suitable form.

Minor comments:

  1. unify the way numbers are presented (one decimal point or two if needed in consecutive way): e.g. '8.0' instead of '8'
  2. Figure 1 is unreadable - names of aminoacides are overwritten
  3. The most important features in bread proximate composition and texture analysis for samples D,F and B should be compared and presented in graphical form. It is hard for the reader go through the numbers and description (section 3.2) to identify differences in oarticular features for the final sets of bread

If modification mentioned above are applied, I can recommend the paper for possible publication in Foods journal

Reviewer 3 Report

The Authors had improved a bit the manuscript however it seems that they are not specialized in bread as some important methodological errors were done.

Please do not insert any parameters that can be calculated from TPA tests, read the theory first as Gumminess is mutually exclusive with Chewiness since a product would not be both a semi-solid and solid at the same time.

The same is with Adhesion which is not recommended to measure during TPA test. It needs other accessories.

Remove the unnecessary data to redo the statistics and conclusions.

Some editorial stuff:

Line 259: Total Titratable acidity - lower case letter for titratable

Line 495 - 502 and line 508-515 and line 520-530 remove spacing between the A-H description and locate appropriately the Figures to be easily read. 

Generally, please edit better the tables and Figure location to avoid partitioning of tables or figures and figures caption on different pages. Such an edition seems clumsy and disrespectful for the readers.

Table 6 is another example - minimize the font size - Foods are fully digital journal and any page can be enlarged at the monitor - but an incorrectly prepared table looks bad in any magnification.
